

# Groundwater withdrawal in randomly heterogeneous coastal aquifers

Martina Siena[1], Monica Riva[1,2]

[1]Dipartimento di Ingegneria Civile e Ambientale, Politecnico di Milano, Piazza L. Da Vinci 32, 20133 Milano, Italy.
5 [2]Department of Hydrology and Atmospheric Sciences, University of Arizona, Tucson, AZ 85721, USA.

*Correspondence to*: Martina Siena (martina.siena@polimi.it)

**Abstract.** We analyze the combined effects of aquifer heterogeneity and pumping operations on seawater intrusion (SWI) in coastal aquifers, a phenomenon which is threatening Mediterranean and worldwide regions. We conceptualize the aquifer as a three-dimensional randomly heterogeneous porous medium, where the spatial distribution of permeability is uncertain. The 10 geological setting of our study is patterned after the coastal aquifer of the Argentona river basin, in the Maresme region of Catalonia (Spain). Numerical simulations of transient, three-dimensional, variable-density flow and solute transport are performed within a stochastic Monte Carlo framework. We consider a variety of groundwater withdrawal schemes, designed by varying the screen location along the vertical direction and the distance of the wellbore from the coastline and from the freshwater-saltwater mixing zone, in order to assess the impact of the pumping scenario on the contamination of the 15 freshwater pumping well for a prescribed production rate. SWI is analyzed by examining isoconcentration curves and global dimensionless quantities characterizing (*i*) inland penetration of the saltwater wedge and (*ii*) width of the mixing zone. Our results indicate that heterogeneity affects the (three-dimensional) seawater wedge either in the presence or in the absence of pumping, by reducing toe penetration and enlarging the width of the mixing zone. Simultaneous extraction of fresh and saltwater from two screens along the same wellbore located within the transition zone is effective in limiting SWI during 20 groundwater resources exploitation.

## 1 Introduction

Groundwater resources in worldwide coastal aquifers are seriously threatened by seawater intrusion (SWI) phenomena, which can deteriorate the quality of freshwater aquifers, thus limiting their potential use. This situation is particularly exacerbated within areas associated with intense anthropogenic activities, in connection with high water demands for civil, 25 agricultural and/or industrial processes. Highly critical scenarios are associated with SWI reaching extraction wells designed for urban freshwater supply, with severe environmental, social and economic implications (e.g., Custodio, 2010; Post and Abarca, 2010; Mas Pla et al., 2014; Mazi et al., 2014).

The development of effective strategies for sustainable use of groundwater resources in coastal regions is subordinated to a comprehensive understanding of SWI phenomena. This challenging problem has been originally studied by




assuming a static equilibrium between freshwater (FW) and seawater (SW) and a sharp FW-SW interface, where FW and SW are considered as immiscible fluids. Under these hypotheses, the vertical position of the FW-SW interface below the sea level, $z$, is given by the Ghyben-Herzberg solution (Ghyben, 1888; Herzberg, 1901), $z = h_F \, \rho_F / \Delta\rho$, where $h_F$ is the FW head above the sea level and $\Delta\rho = \rho_S - \rho_F$ is the density contrast, $\rho_S$ and $\rho_F$ respectively being SW and FW density.

Starting from these works, several analytical or semi-analytical expressions have been developed to describe SWI in diverse flow configurations (e.g., Strack, 1976; Dagan and Zeitoun, 1998; Bruggeman 1999; Cheng et al., 2000; Bakker 2003, 2006; Nordbotten and Celia, 2006; Park et al. 2009). In this broad context, Strack (1976) evaluated the maximum (or critical) pumping rate to avoid encroachment of SW in pumping wells used for FW supply. All of these sharp-interface based solutions neglect a key aspect of SWI phenomena, i.e., the formation of a transition zone where mixing between FW and SW

takes place and fluid density varies with salt concentration. As a consequence, these models may significantly overestimate the actual penetration length of the SW wedge, leading to an excessively conservative evaluation of the critical pumping rate (Gingerich and Voss, 2005; Zhou et al., 2000; Pool and Carrera, 2011; Llopis-Albert and Pulido-Velazquez, 2013).

   A more realistic approach relies on the formulation and solution of a variable-density problem, in which SW and FW are considered as miscible fluids and groundwater density depends on salt concentration. The complexity of the problem

typically prevents its solution via analytical or semi-analytical methods, with few notable exceptions (Henry, 1964; Dentz et al., 2006; Bolster et al. 2007; Zidane et al., 2012; Fahs et al. 2014). Henry (1964) developed a semi-analytical solution for a variable-density diffusion problem in a (vertical) two-dimensional homogeneous and isotropic domain. Dentz et al. (2006) and Bolster et al. (2007) applied perturbation techniques to solve analytically the Henry problem for a range of (small and intermediate) values of the Péclet number, which characterizes the relative strength of convective and dispersive transport

mechanisms. Zidane et al. (2012) solved the Henry problem for realistic (small) diffusion coefficients. Fahs et al. (2014) presented a semi-analytical solution for a system characterized by a square porous cavity subject to different salt concentrations at its vertical walls. The use of these solutions in practical applications is quite limited, mainly because the effects of mechanical dispersion are neglected. A variety of numerical codes have been proposed to solve variable-density flow and transport equations (e.g., Voss and Provost, 2002; Ackerer et al., 2004; Soto Meca et al., 2007; Ackerer and

Younes, 2008; Albets-Chico and Kassinos, 2013). Abarca et al. (2007) introduced a modified Henry problem to account for dispersive solute transport and anisotropy in hydraulic conductivity. Kerrou and Renard (2010) analyzed the dispersive Henry problem within two- and three-dimensional randomly heterogeneous aquifers. These authors rely on computational analyses performed on a single three-dimensional realization, invoking ergodic assumptions. Lu et al. (2013) performed a set of laboratory experiments and numerical simulations to investigate the effect of geological stratification on SW-FW mixing.

Riva et al. (2015) considered the same setting as in Abarca et al. (2007) and studied the way quantification of uncertainty associated with SWI features is influenced by lack of knowledge of four key dimensionless parameters controlling the process, i.e. gravity number, permeability anisotropy ratio and transverse and longitudinal Péclet numbers. Enhancement of mixing in the presence of tidal fluctuations and/or FW table oscillations has been analyzed by Ataie-Ashtiani et al. (1999),





Lu et al. (2009), Pool et al. (2014) and Lu et al. (2015) in homogeneous aquifers and by Pool et al. (2015) in randomly heterogeneous three-dimensional systems (under ergodic conditions). A recent review on this topic is offered by Ketabchi et al. (2016).

Several variable-density numerical models have been developed to identify the most effective strategy for the exploitation of groundwater resources in (homogeneous or heterogeneous) deterministic systems, mimicking the behavior of specific field sites. Narayan et al. (2007) employed a two-dimensional vertical model to assess the potential role of artificial recharge in controlling SWI in the Burdekin deltaic region of North Queensland (Australia). Misut and Voss (2007) analyzed the impact of aquifer storage and recovery practices on the transition zone associated with the salt water wedge in the New York City aquifer, which was modeled as a three-dimensional perfectly stratified system. Cobaner et al. (2012) studied the effect of transient pumping rates from multiple wells on SWI in the Gosku deltaic plain (Turkey) by means of a three-dimensional heterogeneous model, calibrated using head and salinity data. Dougeris and Zissis (2014) developed a three-dimensional finite-element numerical model to simulate variable-density flow and evaluate the quality of groundwater extracted from a synthetic coastal aquifer in the presence of diverse steady-state pumping schemes.

The effect of groundwater withdrawal on SWI within a randomly heterogeneous system has been analyzed by Koussis et al. (2002) for a two-dimensional domain representative of a vertical section of the Rhodes aquifer. The authors inspected the benefit of combining groundwater withdrawal with wastewater recharge on controlling the extent of SWI phenomena.

Here, we study the features of a SWI scenario considering a variable-density flow taking place in a three-dimensional randomly heterogeneous aquifer. We frame our analysis within a numerical Monte Carlo approach and analyze the joint effects of heterogeneity and groundwater withdrawals on the strength of SWI. The latter is here quantified through the study of general spatial pattern of salt concentration distributions and global dimensionless quantities characterizing the SW wedge and quantifying the extent of (*i*) inland penetration and (*ii*) width of the mixing zone. All of these quantities are evaluated within each heterogeneous aquifer realization and by relying on the ensemble averaged concentration field. The latter includes the uncertainty in the displacement of the center of mass of the solute distribution as well as solute spreading (Dentz and Carrera, 2005 and reference therein). Therefore, our study also contributes to identifying benefits and limitations of methods where SWI features are directly inferred from average concentration fields that are typically estimated through interpolation of available concentration data (e.g., Rivest et al., 2012; Lu et al., 2016).

Our numerical model has been designed to mimic the general behavior of the Argentona aquifer, in the Maresme region of Catalonia (Spain). This area, as well as other Mediterranean deltaic sites, is particularly vulnerable to SWI (Custodio, 2010). A review of SWI phenomena in several deltaic regions of Catalonia has been presented by Mas-Pla et al. (2014), who also provided an overview of the technologies applied to deal with (or possibly prevent) SWI. One of these techniques, namely double-negative hydraulic barrier, relies on the joint use of a FW (*production* well) and a SW (*scavenger* well) pumping well. The effectiveness of simultaneous FW and SW pumping in preventing SWI into homogeneous aquifers has been investigated by Aliewi et al. (2001), Pool and Carrera (2009) and Saravanan et al. (2014). These studies show that



the efficiency of a double-negative hydraulic barrier can be improved by (*i*) moving the SW well close to the sea, (*ii*) limiting the screen of the production and scavenger wells respectively at the top and the bottom of the aquifer, (*iii*) reducing the horizontal distance between production and scavenger wells, so that the upconing due to FW pumping is balanced by the downconing due to SW extraction. Aliewi et al. (2001) and Saravanan et al. (2014) observed that optimal conditions are obtained when the pumping rate of the scavenger well is greater than or equal to half the pumping rate of the production well. In our study, we analyze the impact of single and double-negative barriers located at diverse distances from the shoreline on SWI features. Our double-negative barrier configurations are designed by locating production and scavenger wells along the same wellbore, complying with technical and economical requirements typically associated with field applications. The analyzed scheme is also particularly appealing when applied to renewable energy resources, such as the Pressure Retarded Osmosis, that allows converting the chemical energy of two fluids (FW and SW) into mechanical and electrical energy (Panyor, 2006).

The work is organized as follows. Section 2 provides a general description of the field site, the mathematical model adopted to simulate flow and transport phenomena in three-dimensional heterogeneous systems and the numerical settings. Section 3 illustrates the key results of our work. Section 4 contains our concluding remarks.

## 2 Materials and methods

### 2.1 Site description

To consider a realistic scenario that is relevant to SWI problems in highly exploited aquifers, we cast our analysis in a setting inspired from the Argentona river basin, located in the Catalan region of Maresme (see Fig. 1a). This aquifer is a typical deltaic site, characterized by shallow sedimentary units and a flat topography. As such, it has a strategic value for anthropogenic (including agricultural, industrial and touristic) activities. The geological formation hosting the groundwater resource is mainly composed by a granitic Permian unit. A secondary unit of quaternary sediments is concentrated along the Argentona river.

A preliminary site characterization and water-balance study of the aquifer has been performed by the Agència Catalana de l'Aigua (ACA), the local authority for water resources. Almagro Landò et al. (2010) developed a conceptual and numerical model to simulate transient two-dimensional (horizontal) constant-density flow in the Argentona river basin across the area of about 35 km² depicted in Fig. 1a. The aquifer is heavily exploited, as shown by the large number of withdrawal wells included in Fig. 1a, mainly located along the Argentona river. The latter is a torrential ephemeral stream, in which water flows only after heavy rain events. On these bases, Almagro Landò et al. (2010) assumed that the only water intake to the river comes from surface runoff of both granitic and quaternary units. The conceptual and numerical model of Almagro Landò et al. (2010) includes information about geological units, hydrological properties, as well as monthly data of groundwater extractions and natural recharge over a 4-year period, i.e., 2006-2009. The model has been recently updated by Rodriguez Fernandez (2015) including groundwater pumping rates collected from January 2010 to December 2013. On the





basis of transient hydraulic head measurements available at 21 observation wells (not shown in Fig. 1a), Almagro Landò et al. (2010) and Rodriguez Fernandez (2015) characterized the site through a uniform permeability whose value was estimated as $k_B = 1.77 \times 10^{-11} \, \text{m}^2$. These authors also provide estimates of temporally and spatially variable recharge rates, according to land use or cover.

Our numerical analysis focuses on the coastal portion of the Argentona basin (Fig. 1b). This region extends for about 2.5 km along the coast (i.e. along the full width of the basin) and up to 750 m inland from the coast, where SWI phenomena are expected to be relevant. It has a vertical thickness of 50 m, the underlying clay acting as impervious boundary. SWI is simulated within this domain by means of a three-dimensional variable-density flow and solute transport model embedded in the finite element USGS SUTRA code (Voss and Provost, 2002) over the 8-year time window 2006 - 2013. Details of the
mathematical and numerical model are discussed in the following Sections. Table 1 lists aquifer and fluids parameters adopted.

## 2.2 Flow and transport equations

Fluid flow is governed by mass conservation and Darcy's Law:

$$\frac{\partial(\phi\rho)}{\partial t} + \nabla^T(\rho \boldsymbol{q}) = 0 \qquad \text{with} \qquad \frac{\partial(\phi\rho)}{\partial t} = \frac{1}{g} S_S \frac{\partial p}{\partial t} + \phi \frac{\partial \rho}{\partial C} \frac{\partial C}{\partial t} \qquad (1)$$

$$\boldsymbol{q} = -\frac{\boldsymbol{k}}{\mu}(\nabla p + \rho g \nabla z) \qquad (2)$$

where $\boldsymbol{q}$ [L T$^{-1}$] is the specific discharge vector, with components $q_x$, $q_y$ and $q_z$ respectively along $x$-, $y$- (see Fig. 1b) and $z$-axis (vertical direction), $C$ [-] is solute concentration, or solute mass fraction, expressed as mass of solute per unit mass of fluid, $\boldsymbol{k}$ [L$^2$] is the diagonal permeability tensor, with components $k_{11} = k_x$, $k_{22} = k_y$ and $k_{33} = k_z$, respectively along directions $x$, $y$ and $z$, $\phi$ [ - ] is aquifer porosity, $\rho$ [M L$^{-3}$] and $\mu$ [M L$^{-1}$ T$^{-1}$] are fluid density and dynamic viscosity, $p$
[M L$^{-1}$ T$^{-2}$] is fluid pressure, $g$ [L T$^{-2}$] is gravity, $S_S$ [L$^{-1}$] is the specific storage coefficient, and the superscript $T$ denotes transpose. Equations (1) and (2) must be solved jointly with the advection-dispersion equation:

$$\frac{\partial(\phi\rho C)}{\partial t} + \nabla^T(\rho C \boldsymbol{q}) - \nabla^T[\rho \boldsymbol{D} \nabla C] = 0 \qquad (3)$$

$\boldsymbol{D}$ [L$^2$ T$^{-1}$] being the dispersion tensor defined as

$$\boldsymbol{D} = (\phi D_m + \alpha_T |\boldsymbol{q}|)\mathbf{I} + (\alpha_L - \alpha_T)\frac{\boldsymbol{q}\boldsymbol{q}^T}{|\boldsymbol{q}|} \qquad (4)$$





where $D_m$ is the molecular diffusion coefficient, and $\alpha_L$ and $\alpha_T$ [L] respectively are the longitudinal and transverse dispersivity coefficients. Closure of system of Eqs. (1)-(3) requires a relationship between fluid properties ( $\rho$ and $\mu$ ) and solute concentration $C$ . Fluid viscosity can be assumed constant in typical SWI settings and the following model has been shown to be accurate to describe the evolution of $\rho$ with $C$ (Kolditz et al., 1998)

$$\rho = \rho_F + (\rho_S - \rho_F)\frac{C}{C_S} \tag{5}$$

where $\rho_F$ and $\rho_S$ respectively are FW and SW densities, and $C_S$ is SW concentration.

## 2.3 Numerical model

The domain depicted in Fig. 1b is discretized through an unstructured three-dimensional grid formed by 101632 hexahedral elements. The resolution of the mesh increases towards the sea, where our analysis requires the highest spatial detail. The element size along both horizontal directions ranges between 60 m (close to the inland boundary) to 10 m (in a 200m-wide region along the coast). Element size along the vertical direction is 2 m and 4 m, respectively within the first 10 m from the top surface and across the remaining 40 m. These choices are consistent with constraints for numerical stability, namely $\Lambda_L \leq 4\alpha_L$ , $\Lambda_L$ being the distance between element sides measured along a flow line; here, we set $\alpha_L$ = 5 m (see also Table 1), consistent with the estimate provided by Almagro Landò et al. (2010). In the absence of transverse dispersivity estimates and in analogy with main findings of previous works (e.g., Cobaner et al., 2012), we set $\alpha_T = \alpha_L/10 = 0.5$ m. The model (see Fig. 1) is laterally bounded by topographic divides that are associated with no-flow conditions, these being imposed also at the bottom of the aquifer. Vertical profiles of stationary hydrostatic pressure are set along the coastline. Transient boundary conditions of prescribed head and flux are respectively set along the inland boundary and at the top of the aquifer. These have been obtained by using head and recharge values provided by Almagro Landò et al. (2010) and Rodriguez Fernandez (2015), as discussed in Sect. 2.1. Solute concentration in the fluid entering the domain along the inland (north-west) boundary and from the top of the aquifer is set equal to FW concentration, $C_F$ . Water entering and leaving the system across the coastal boundary has concentration respectively equal to $C_S$ and $C$ (the latter concentration being unknown and obtained as the numerical solution of the problem). This condition is set as:

$$(\boldsymbol{q}C - \boldsymbol{D}\nabla C)\cdot\boldsymbol{n} = \begin{cases} q_y C & if \ q_y < 0 \\ q_y C_S & if \ q_y > 0 \end{cases} \tag{6}$$

$\boldsymbol{n}$ being the normal vector pointing inward along the boundary.

A Sequential Gaussian simulation algorithm (Deutsch and Journel, 1998) is employed to generate 60 unconditional realizations of $Y(\boldsymbol{x}) = \ln k(\boldsymbol{x})$ characterized by a given mean $\langle Y \rangle = \ln k_B$ and variogram structure. Since no $Y$ data are





available, for the purpose of our simulations we assume that the spatial structure of $Y$ is described by a spherical variogram, with moderate variance, i.e., $\sigma_Y^2 = 1.0$, and isotropic correlation scale $\lambda = 100$ m. These hypotheses are consistent with the observation that the integral scale of log conductivity and transmissivity values inferred worldwide using traditional (such as exponential and spherical) variograms tends to increase with the length scale of the sampling window at a rate of

about 1/10 (Gelhar, 1993; Neuman et al., 2008). We remark that the objective of this study is not the quantification of the SWI dynamics of a particular field site. Our emphasis is on the analysis of the impacts of random aquifer heterogeneity and withdrawals on SWI patterns in a realistic scenario. Note also that $\lambda > 5\Delta$ in our model, $\Delta$ being the largest cell size within a 200 m-wide region along the coast. A model satisfying this condition has the benefit of (*a*) yielding a good reconstruction of the correlation structure of $Y$ (Ababou et al., 1989) and (*b*) limiting the occurrence of excessive variations

between values of aquifer properties across neighboring cells (Kerrou and Renard, 2010) in the region where SWI occurs (see Sect. 3).

## 3 Results and discussion

### 3.1 Effects of three-dimensional heterogeneity on SWI

Numerical simulations of transient flow and transport spanning over an 8-year time window (2006-2013) are performed on

the collection of the 60 heterogeneous realizations of $Y$ described in Sect. 2.3. The effects of heterogeneity on SWI are inferred by comparing the results of our Monte Carlo simulations against those obtained for an equivalent homogeneous aquifer, characterized by an effective permeability, $k_{ef}$. The latter is here evaluated as $k_{ef} = e^{\langle Y \rangle + \sigma_Y^2/6} = 2.09 \times 10^{-11} \text{m}^2$ (Ababou, 1996).

A first clear effect of heterogeneity is noted on the structure of the three-dimensional flow field. This is elucidated by Fig. 2,

where we depict permeability color map and streamlines (dashed curves) obtained at the end of the simulation period at the vertical cross-section B-B' perpendicular to the coastline for (*i*) the equivalent homogeneous aquifer with $k = k_{ef}$ (Fig. 2b) and (*ii*) a selected random heterogeneous realization of $k$ (see Fig. 2c). Note that here and in the following, results are depicted in terms of dimensionless spatial coordinates, $y' = y/\lambda$ and $z' = z/\lambda$. Flow within the homogeneous aquifer is essentially horizontal at locations far from the zone where SWI phenomena occur. Otherwise, the vertical flux component

$q_z$ is non-negligible throughout the whole domain for the heterogeneous system and streamlines tend to focus towards regions characterized by large permeability values. Fig. 2 also depicts isoconcentration curves $C/C_S = 0.25, 0.5$ and $0.75$ within the transition zone (red curves). It can be noted that isoconcentration profiles tend to be sub-parallel to streamlines directed towards the seaside boundary. In the homogeneous domain (Fig. 2b) the slope of these curves varies mildly and in a gradual manner from the top to the bottom of the aquifer. Otherwise, their slope in the heterogeneous domain (Fig. 2c) is



irregular and markedly influenced by the spatial arrangement of permeability. In other words, the way streamlines are refracted at the boundary between two blocks of contrasting permeability drives the local pattern of concentration contour-lines. As a consequence, isoconcentration curves tend to become sub-vertical when solute is transitioning from regions characterized by high $k$ values to zones associated with moderate to small $k$, a sub-horizontal pattern being observed when

transitioning from low to high $k$ values.

Figure 3 depicts isolines $C/C_S = 0.5$ at the bottom of the aquifer (Fig. 3a) and along three vertical cross-sections (Figs 3b-3d, selected to exemplify the general pattern observed in the system) evaluated for (*i*) the 60 heterogeneous realizations analyzed (dotted blue curves), (*ii*) the equivalent homogeneous system (denoted as *Hom*; solid red curve) and (*iii*) the configuration obtained by averaging the concentration fields across all Monte Carlo heterogeneous realizations (denoted as

*Ens*; solid blue curve). These results suggest that isoconcentration curves exhibit considerably large spatial variations within a single realization. Moreover, comparison of the results obtained for *Ens* and *Hom* reveals that the mean wedge penetration at the bottom layer is slightly overestimated by the solution computed within *Hom*. Otherwise, along the coastal vertical boundary the extent of the area with (mean) relative concentration larger than 0.5 is generally underestimated by *Hom*. These outcomes (hereafter called rotation effects) are consistent with previous literature findings. In particular, Kerrou and Renard

(2010) and Pool et al. (2015) observed that the strength of the rotation effect increases with the variance of the log-permeability field. Abarca (2006, 2007) showed that a similar effect can be observed in a homogenous domain when considering increasing values of the dispersion coefficient.

Here, we quantify the extent of the SW wedge on the basis of a set of seven quantities. We evaluate for each Monte Carlo realization and along each vertical cross-section perpendicular to the coast: (*i*) toe penetration, $L_T'$, measured as the distance

from the coast of the isoline $C/C_S = 0.5$ at the bottom of the aquifer; (*ii*) solute spreading at the toe, $L_S'$, evaluated as the separation distance along *y*-axis between the isolines $C/C_S = 0.25$ and $C/C_S = 0.75$ at the bottom of the aquifer; (*iii*) mean width of the mixing zone, $W_{MZ}'$, evaluated as the spatially-averaged vertical distance between isolines $C/C_S = 0.25$ and $C/C_S = 0.75$ within the region $0.2\,L_T' \leq y' \leq 0.8\,L_T'$. Note that $L_T'$, $L_S'$ and $W_{MZ}'$ are dimensionless quantities, all of them being distances rescaled by $\lambda$, the correlation scale of *Y*. We also analyze the dimensionless areal extent of SW

penetration and solute spreading at the bottom of the aquifer by computing respectively (*iv*) $A_T'$ and (*v*) $A_S'$, i.e., the integrals of $L_T'$ and $L_S'$ all along the coastline. We further characterize SWI on the whole thickness of the aquifer by evaluating the dimensionless volumes enclosed between (*vi*) the sea boundary and the isoconcentration surface $C/C_S = 0.5$, $V_T'$, and (*vii*) the isosurfaces $C/C_S = 0.25$ and $C/C_S = 0.75, V_S'$.



To highlight heterogeneity effects, we report in the following values of $\xi'' = \xi'/\xi'^{Hom}$, $\xi'$ corresponding to each of the seven quantities listed above and $\xi'^{Hom}$ being its counterpart obtained on the equivalent homogeneous system. Fig. 4 depicts $\xi''$ computed in each heterogeneous realizations (symbols). Each plot is complemented by the depiction of (*i*) the (ensemble) average of $\xi''$ assessed across all Monte Carlo realizations, $\langle \xi'' \rangle$, (black solid line), (*ii*) the confidence

intervals, $\langle \xi'' \rangle \pm \sigma_{\xi''}$, $\sigma_{\xi''}$ being the standard deviation of $\xi''$ (black dashed line) and, (*iii*) $\xi''^{Ens}$, evaluated on the basis of the ensemble averaged concentration field (blue line). Inspection of Figs.4a-4c complements the qualitative analysis of Fig. 3 and suggests that heterogeneity causes (on average) a slight reduction of the toe penetration, $\langle L_T'' \rangle$ and $L_T''^{Ens}$ being slightly smaller than 1, and an enlargement of the mixing zone, $\langle L_S'' \rangle$ and $\langle W_{MZ}'' \rangle$ being larger than 1. The results of Figs. 4a-4c also emphasize that, while $L_T''^{Ens}$ calculated from the ensemble averaged concentration distributions is virtually

indistinguishable from $\langle L_T'' \rangle$, $L_S''^{Ens}$ and $W_{MZ}''^{Ens}$ markedly overestimate $\langle L_S'' \rangle$ and $\langle W_{MZ}'' \rangle$, as they visibly lie outside of the corresponding confidence intervals of width $\pm \sigma_{\xi''}$. Note that, even as Figs. 4 a-c have been computed along cross-section B-B', qualitatively similar results have been obtained along all vertical cross-sections as also suggested by the behavior of the dimensionless areal extent of toe penetration, $A_T''$ (Fig. 4d), and solute spreading, $A_S''$ (Fig. 4e), as well as of the dimensionless volumes $V_T''$ and $V_S''$ (Figs. 4f-4g). Heterogeneity effects decrease whenever integral, rather than local

quantities are considered. Note that $V_S''^{Ens}$ in Fig. 4g is about half of $L_S''^{Ens}$ and $A_S''^{Ens}$. Overall, the results in Fig. 4 clearly indicate that the ensemble averaged concentration field can provide accurate estimates of the wedge penetration while rendering biased estimates of quantities characterizing mixing. Our findings are consistent with previous studies (e.g. Pool et al., 2015) in showing that an analysis relying on the ensemble concentration field tends to overestimate significantly the degree of mixing and spreading of the solute.

**3.2 Effects of pumping on SWI in three dimensional heterogeneous media**

We now investigate SWI phenomena in the heterogeneous systems described in the previous section when a pumping well is located close to the coast. Pumping is activated for 8 months after the 8-year period analyzed in Sect. 3.1. We compare the impact of three diverse pumping schemes, reflecting realistic engineered operational settings, on the seven dimensionless quantities introduced in Sect. 3.1. The borehole is located along the vertical cross-section B-B' (see Fig. 3a). In Scheme 1

(S1) and Scheme 2 (S2) the pumping well is placed 180 m away from the shoreline, i.e. at $y' = 1.8$, outside the transition zones depicted in Fig. 3c. In S1 the well is screened in the upper part of the aquifer, the screen starting from 2 m below the top and extending across a total thickness of 15 m, as illustrated in Figs. 5c-5d. Scheme S2 (see Figs. 5e-5f) is designed by adding to S1 an additional screen along the same wellbore. This screen is located in the lower part of the aquifer, starting



from 34 m below the ground surface and extending across a total thickness of 12 m. Scheme 3 (S3) shares the same operational design of S2, but the borehole is moved seawards, at $y' = 0.8$ (see Fig. 5g-5h), within the transition zone before pumping. The comparison of these schemes addresses a problem of practical interest: in the presence of a FW withdrawal at a constant rate $Q$, we investigate the extent at which the simultaneous extraction of SW at the same rate $Q$ may contribute to

limit SWI. For the whole simulated pumping period, we set $Q = 5$ l/s at the upper well screen in all schemes, i.e., S1-S3; an additional extraction at rate $Q$ is imposed along the lower well screen in schemes S2 and S3. All schemes are compared against a benchmark case, hereafter termed Scheme 0 (S0), where there are no pumping wells.

Figure 5 collects contour maps of relative concentration $C/C_S$ obtained for all schemes along cross-section B-B' at the end of the above illustrated 8-month period. Results for each case are depicted with reference to (*i*) *Hom* (the equivalent

homogeneous system, left column) and (*ii*) *Ens* (the ensemble-averaged concentration field, right column). Contour lines $C/C_S = 0.25$, 0.50 and 0.75 are also highlighted. Extracting FW according to the engineered solution S1 results in a slight landward displacement of the SWI wedge and in an enlargement of the transition zone, as compared to S0. This behavior can be observed in the homogeneous as well as (on average) in the heterogeneous setting and is related to the general decrease of the piezometric head within the inland side caused by pumping which, in turn, favors SWI. One can also note that the

partially penetrating well induces non-horizontal flow (in the proximity of the well) thus enhancing mixing along the vertical direction. Wedge penetration and solute spreading are further enlarged in S2 (see Figs. 5e-5f) where the total extracted volume is increased with respect to S1 through an additional pumping rate at the bottom of the aquifer. However, when the pumping well is operating within the transition zone (scheme S3, Figs. 5g-5h) the SW wedge tends to recede and to be focused around the pumping well location. In this case, the lower screen acts as a barrier limiting the extent of the SWI at the

bottom of the aquifer.

The combined effects of groundwater withdrawal and heterogeneity on SWI are investigated quantitatively through the analysis of the temporal evolution of the seven dimensionless quantities introduced in Sect. 3.1. Figure 6 illustrates mean values of $\xi'$ (with $\xi' = L'_T$, $L'_S$ and $W'_{MZ}$) computed for all pumping schemes considered across the collection of all heterogeneous realizations (black line) along the vertical cross-section B-B' versus dimensionless time $t' = (t - t_w) Q / \lambda^3$,

$t_w$ being the time corresponding to the activation of the withdrawal in S1-S3. On each plot, uncertainty intervals $\langle \xi' \rangle \pm \sigma_{\xi'}$ are also depicted (dashed lines). As additional terms of comparison, Fig. 6 also includes $\xi'^{Hom}$ (red line) and $\xi'^{Ens}$ (blue line), respectively evaluated on the equivalent homogeneous domain and on the ensemble averaged concentration distribution. Fig. 6 shows that the toe penetration, $L'_T$, and the solute spreading, as quantified by $L'_S$ and $W'_{MZ}$, do not vary significantly with time in the absence of pumping (S0) because stationary boundary conditions are imposed for $t' > 0$.

Pumping schemes S1 and S2 cause the progressive inland displacement of the toe, together with an overall increase of spreading at the bottom of the aquifer. This phenomenon is more severe in S2, where SW and FW are simultaneously





extracted. Otherwise, the vertical width of the mixing-zone, e.g. $W'_{MZ}$, is not significantly affected by the pumping within schemes S1-S2. Configuration S3, in which the pumping well is located within the mixing zone, leads to the most pronounced changes in the shape and position of the SW wedge. The toe penetration first decreases rapidly in time and then stabilizes around the well location. Quantities $L'_S$ and $W'_{MZ}$, which are representative of the width of the transition zone,

show an early-time increase, suggesting that a rapid displacement of the wedge enhances mixing and spreading. As the toe stabilizes over time, both $L'_S$ and $W'_{MZ}$ decrease, reaching values equal to (or slightly smaller than) those detected in the absence of pumping. Consistent with the observations of Section 3.1, Fig. 6 supports the findings that (*i*) an equivalent homogeneous domain is typically characterized by the largest toe penetration and the smallest vertical width of the mixing zone, the only exception being observed in S3 at late times, where $L'^{Hom}_T \cong L'^{Ens}_T < \langle L'_T \rangle$ ; (*ii*) grounding the

characterization of the mixing-zone on ensemble averaged concentrations enhances the actual effects of heterogeneity and yields overestimated mixing-zone widths. Moreover, Fig. 6 shows that the extent of the confidence interval associated with toe penetration is approximately constant in time and does not depend significantly on the analyzed flow configuration. Otherwise, the confidence intervals associated with mixing-zone parameters depend on the pumping scenario and tend to increase with time in schemes S2 and S3.

The fully three-dimensional nature of the analyzed problem is exemplified in Fig. 7, where we depict isolines $C/C_S$ = 0.5 at the bottom of the aquifer at the end of the pumping period for S0-S3 for the equivalent homogeneous system (Fig. 7a) and as a result of ensemble averaging across the collection of heterogeneous fields (Fig. 7b). The spatial pattern of isoconcentration contours associated with pumping schemes S1-S3 departs from the corresponding results associated with S0 within a range extending for about 10 $\lambda$ from the well along the direction parallel to the coastline. We quantify the global

effect of pumping on the three-dimensional SW wedge by evaluating, for each pumping scenario and each Monte Carlo realization, the temporal evolution of $\xi^* = 100 \times \left( \xi' - \xi'_{S_0} \right) \big/ \xi'_{S_0}$ , i.e., the relative percentage variation of areal ( $\xi' = A'_T, A'_S$ ) and volumetric ( $\xi' = V'_T, V'_S$ ) extent of wedge penetration and mixing zone with respect to their counterparts computed in the absence of pumping ( $\xi'_{S_0}$ ). Figure 8 shows that the penetration area, $A^*_T$, and the solute spreading, $A^*_S$, at the bottom of the aquifer tend to increase with pumping time. The scenarios where these quantities are

most and least affected by pumping are respectively S2 and S3. We further note that the mean penetration area can be accurately determined by the solution obtained within the equivalent homogeneous domain and is associated by a relatively small uncertainty, as quantified by the confidence intervals of width $\pm\sigma_{A^*_T}$. Heterogeneity effects are clearly visible on spreading, i.e., on $A^*_S$: Figs 8d-f highlight that $A^{*Hom}_S$ and $A^{*Ens}_S$ are not accurate approximations of $\langle A^*_S \rangle$. Spreading





uncertainty, as quantified by $\sigma_{A_S^*}$, is significantly larger than $\sigma_{A_T^*}$, especially in pumping scenario S2. Qualitatively similar conclusions can be drawn by looking at $V_T^*$ and $V_S^*$, in Fig. 9, even as heterogeneity tends to affect more significantly the SWI area at the bottom of the aquifer than the overall SWI volume.

Our analyses document that an operational scheme of the kind engineered in S3 (*i*) is particularly efficient for the

5 reduction of SWI maximum penetration (localized at the bottom of the aquifer) and (*ii*) compared to the double-negative barrier in S2, is advantageous in controlling the extent of the volume of the SW wedge, by limiting it. However, one should also note that this withdrawal system operates within the transition zone and may lead to the salinization of the FW extracted from the upper screen due to upconing effects. This aspect is further analyzed in Fig. 10 where the temporal evolution of $C_T/C_S$ is depicted, $C_T$ being the salt concentration associated with the total mass of fluid extracted by the upper screen

(production well) in S3. Results for the equivalent homogeneous system (red curve), each of the heterogeneous Monte Carlo realizations (grey dashed curves) and the ensuing ensemble averaged result $\langle C_T \rangle / C_S$ (blue curve) are depicted. Vertical bars represent the 95% confidence interval around the ensemble mean, evaluated as $\langle C_T \rangle / C_S \pm 1.96\, \sigma_{C_T/C_S} / \sqrt{n}$, where $\sigma_{C_T/C_S}$ is the standard deviation of $C_T/C_S$ and $n = 60$ is the number of Monte Carlo simulations. The width of these confidence intervals can serve as metric to quantify the order of magnitude of the uncertainty associated with $\langle C_T \rangle / C_S$.

The equivalent homogeneous system significantly underestimates the ratio $\langle C_T \rangle / C_S$. Moreover, Fig. 10 highlights the marked variability of the results across the Monte Carlo space. As such, the mean value $\langle C_T \rangle$ is an intrinsically weak indicator of the actual salt concentration at the producing well.

## 4 Conclusions

We investigate the role of heterogeneity and groundwater withdrawal on seawater intrusion (SWI) in coastal aquifers

through a suite of numerical simulations of transient variable density flow and transport in a three-dimensional system. The latter is conceptualized as a heterogeneous medium whose permeability is a random function of space. In order to consider a realistic and relevant scenario, the numerical model has been tailored to the general hydrogeological setting of the coastal aquifer of the Argentona river basin (Spain), a region which is massively plagued by SWI. The phenomenon is studied through the analysis of (*a*) the general pattern of isoconcentration curves and (*b*) global dimensionless quantities describing

the average toe penetration and the extent of the mixing or transition zone in one ($L_T'$, $L_S'$, $W_{MZ}'$), two ($A_T'$, $A_S'$) and three dimensions ($V_T'$, $V_S'$). We compare results obtained across the collection of heterogeneous aquifer realizations and on an equivalent homogeneous system. With reference to the heterogeneous settings considered, the results based on a



quantification of the SWI process within each realization of solute concentration are compared against those obtained by relying on the (ensemble) averaged concentration field. Our work leads to the following major conclusions.

1. Natural heterogeneity of the system affects the SW wedge along all directions either in the presence or in the absence of pumping. On average, heterogeneous aquifer systems are characterized by toe penetration and extent of the mixing zone that are respectively smaller and larger than their counterparts computed in the equivalent homogeneous system.

2. Values of $L_T'$ $A_T'$, and $V_T'$ calculated on the basis of the ensemble averaged concentration field virtually coincide with their (ensemble) mean counterpart, estimated across the collection of Monte Carlo realizations.

3. Relying on the spatial distribution of ensemble averaged concentrations leads to markedly overestimating the ensemble averages of $W_{MZ}'$, $L_S'$, and $A_S'$ and to slightly overestimating the ensemble average of $V_S'$, the effects of heterogeneity being damped when considering the wedge size associated a three-dimensional setting. On these bases, average concentration fields, typically estimated through interpolation of available concentration data, cannot be employed to provide a reliable estimates of solute spreading and mixing.

4. All of the tested pumping schemes lead to an increased SW wedge volume compared to the scenario where pumping is absent. The key aspect controlling the effects of groundwater withdrawal on SWI is the position of the wellbore with respect to the location of the saltwater wedge prior to pumping. The toe penetration decreases or increases depending on whether the well is initially (i.e., before pumping) located within or outside the seawater intruded region, respectively. The water withdrawal scheme that is most efficient for the reduction of the maximum inland penetration of the seawater toe is the one according to which freshwater and saltwater are respectively extracted from the top and the bottom of the same borehole, initially located within the SW wedge. This scheme is also the most advantageous for constraining the SW wedge volume.

5. Salt concentration, $C_T$, of water pumped from the producing well is strongly affected by permeability heterogeneity. Our Monte Carlo simulations document that $C_T$ can vary by more than two orders of magnitude amongst individual realizations, even in the moderately heterogeneous aquifer considered in this study. As such, relying solely on values of $C_T$ obtained through an effective homogeneous system approximation or ensemble averaged estimates $\langle C_T \rangle$ do not yield a reliable quantification of the actual salt concentration at the producing well.

**Acknowledgments**

This work was supported by MIUR (Italian ministry of Education, Universities and Research) project: ''Hydroelectric energy by osmosis in coastal areas'' (PRIN2010-11) and by project "WE-NEED- Water NEEDs, availability, quality and sustainability" (WaterWorks2014). We are grateful to Albert Folch, Xavier Sanchez-Vila, Laura del Val Alonso of the



Universitat Politècnica de Catalunya and Jesus Carrera of the Spanish Council for Scientific Research for sharing with us data on the hydrogeological characterization of the Argentona site.

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



| Parameter | Value |
|---|---|
| Freshwater density, $\rho_F$ [kg m$^{-3}$] | 1000 |
| Seawater density, $\rho_S$ [kg m$^{-3}$] | 1025 |
| Freshwater mass fraction, $C_F$ [-] | 0.0 |
| Seawater mass fraction, $C_S$ [-] | 0.035 |
| Fluid viscosity, $\mu$ [kg m$^{-1}$ s$^{-1}$] | 0.001 |
| Effective porosity, $\phi$ [-] | 0.15 |
| Specific storage, $S_S$ [m$^{-1}$] | 0.01 |
| Permeability, $k_B$ [m$^2$] | $1.77 \times 10^{-11}$ |
| Molecular diffusion, $D_m$ [m$^2$ s$^{-1}$] | $1 \times 10^{-9}$ |
| Longitudinal dispersivity, $\alpha_L$ [m] | 5.0 |
| Transverse dispersivity, $\alpha_T$ [m] | 0.5 |

**Table 1.** Parameters adopted in the numerical model.





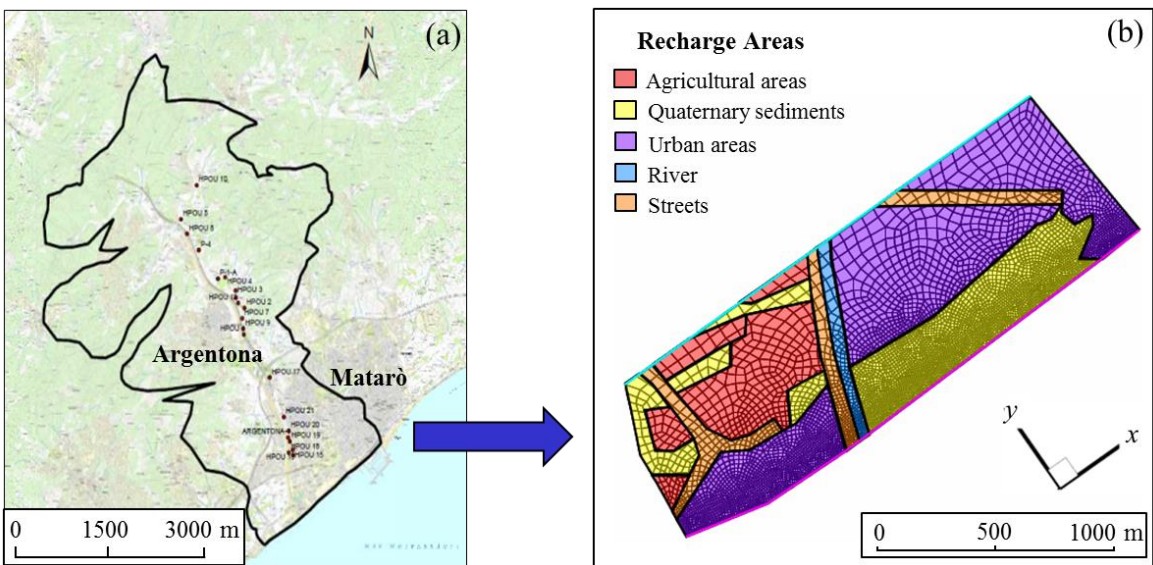

**Fig. 1.** (a) Location of the two-dimensional, constant-density ACA model (modified from Rodriguez Fernandez, 2015); position of pumping wells are also depicted (brown dots). (b) Planar view of the three-dimensional variable-density flow and transport model, unstructured grid and diverse recharge areas are also shown.





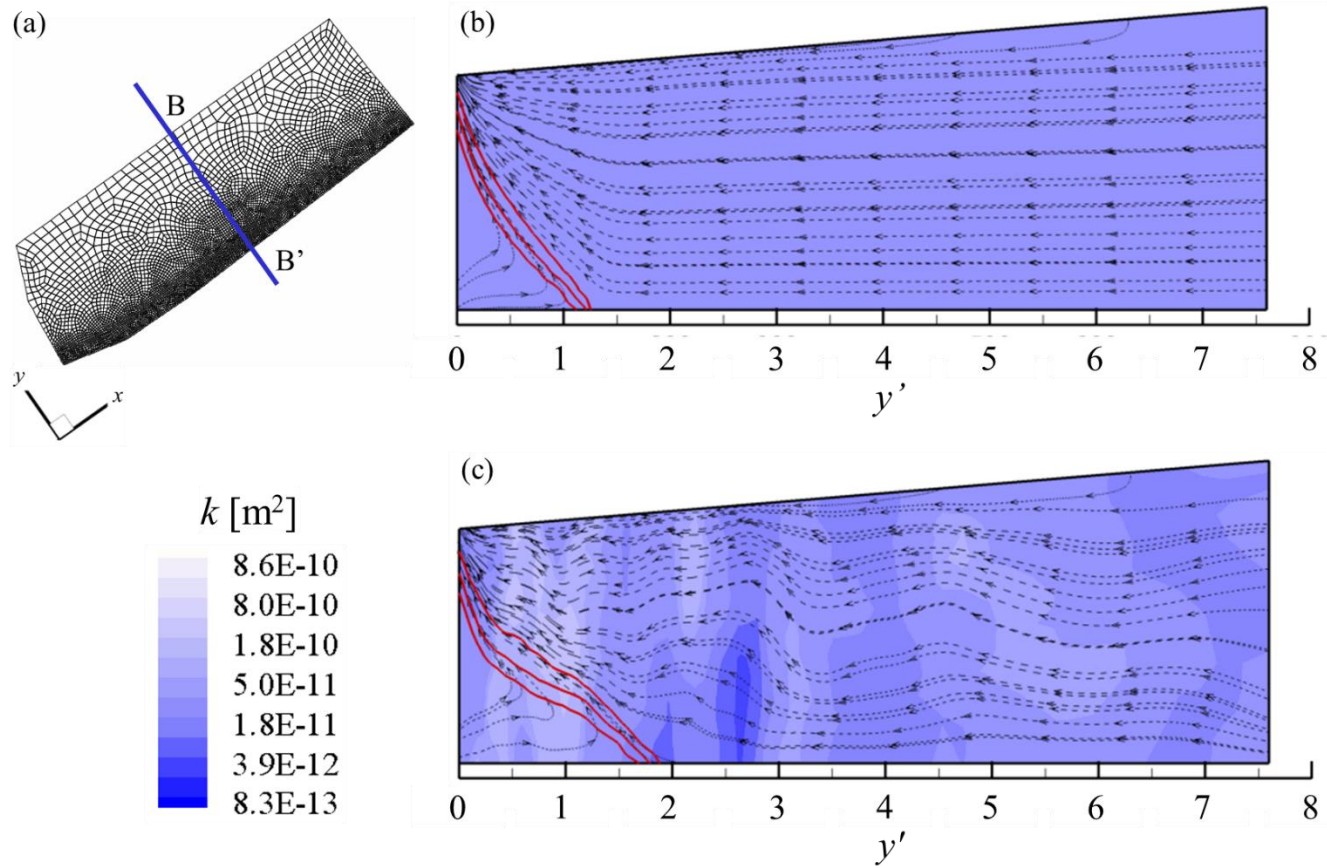

**Fig. 2.** Permeability distribution (color contour plots), streamlines (black dashed curves), isolines $C/C_S = 0.25, 0.5$ and $0.75$ (solid red curves) along the cross-section B-B' highlighted in (a) and evaluated at the end of the 8-year simulated period within (b) the equivalent homogeneous aquifer and (c) one random realization of $k$. Vertical exaggeration = 5.







**Fig. 3.** Isolines $C/C_S = 0.5$ at the end of the 8-year simulated period for the collection of random realizations (dotted blue curves), along (a) the bottom of the aquifer and (b)-(d) three vertical cross-sections perpendicular to the coast. The ensemble average concentration curves (solid blue curve) and the results obtained within the equivalent homogeneous system (solid red curve) are also reported. Vertical exaggeration = 5.



**Fig. 4.** Values of $\xi'' = \xi'/\xi'^{Hom}$ with (a-c) $\xi' = L'_T, L'_S$ and $W'_{MZ}$, evaluated along the vertical cross-section B-B' of Fig. 3, (d-e) $\xi' = A'_T$ and $A'_S$ and (f-g) $\xi' = V'_T$ and $V'_S$ evaluated at the end of the 8-year simulated period within each heterogeneous realization (dots) and on the basis of the ensemble averaged concentration field (blue line). Mean values (solid black line) and confidence intervals of width equal to ± one standard deviation of $\xi''$ about their mean (dashed black lines) are also depicted.





**Fig. 5.** Concentration distribution (color contour plots) and isolines $C/C_S = 0.25, 0.5$ and $0.75$ (solid red curves) along the cross-section B-B' of Fig. 3a evaluated at the end of the pumping period within the equivalent homogeneous aquifer (left column). The ensemble averaged concentration fields obtained from the heterogeneous realizations are also depicted (right column). Black vertical lines represent the location of the well screens. Vertical exaggeration = 5.





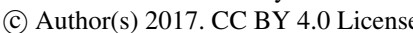

**Fig. 6.** Temporal evolution of $\xi' = L_T', L_S'$ and $W_{MZ}'$ evaluated along the vertical cross-section B-B' of Fig. 3a during the pumping period for all pumping schemes within the equivalent homogeneous domain (red lines) and the ensemble averaged concentration field (blue lines). Monte Carlo-based mean values of $\xi'$ (black lines) and confidence intervals of width equal to $\pm$ one standard deviation of

5   $\xi'$ about their mean are also depicted.





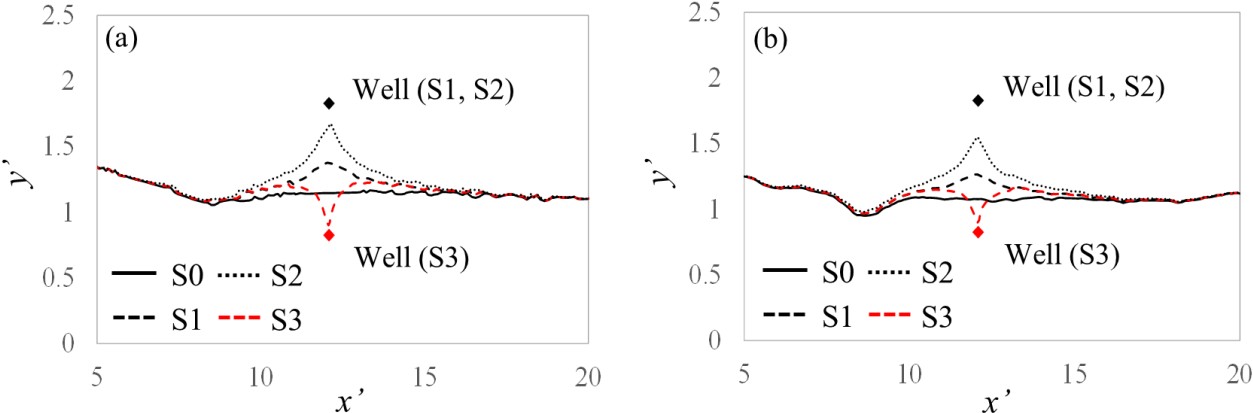

**Fig. 7.** Isolines $C/C_S = 0.5$ at the end of the pumping period along the bottom of the aquifer for the four schemes S0-S3 evaluated (a) within the equivalent homogeneous domain and (b) from the ensemble averaged concentration field.






**Fig. 8.** Temporal evolution of $\xi^* = 100 \times \left( \xi' - \xi'_{S_0} \right) / \xi'_{S_0}$ where $\xi' = A'_T$ and $A'_S$ evaluated for all pumping schemes (S1, S2, S3) within the equivalent homogeneous domain (red lines) and the ensemble averaged concentration field (blue lines). Monte Carlo-based mean values of $\xi^*$ (black lines) and confidence intervals of width equal to $\pm$ one standard deviation of $\xi^*$ about their mean are also depicted.





**Fig. 9.** Temporal evolution of $\xi^* = 100 \times \left( \xi' - \xi'_{S_0} \right) / \xi'_{S_0}$ where $\xi' = V'_T$ and $V'_S$, evaluated for all pumping schemes (S1, S2, S3) within the equivalent homogeneous domain (red lines) and the ensemble averaged concentration field (blue lines). Monte Carlo-based mean values of $\xi^*$ (black lines) and confidence intervals of width equal to ± one standard deviation of $\xi^*$ about their mean are also depicted.



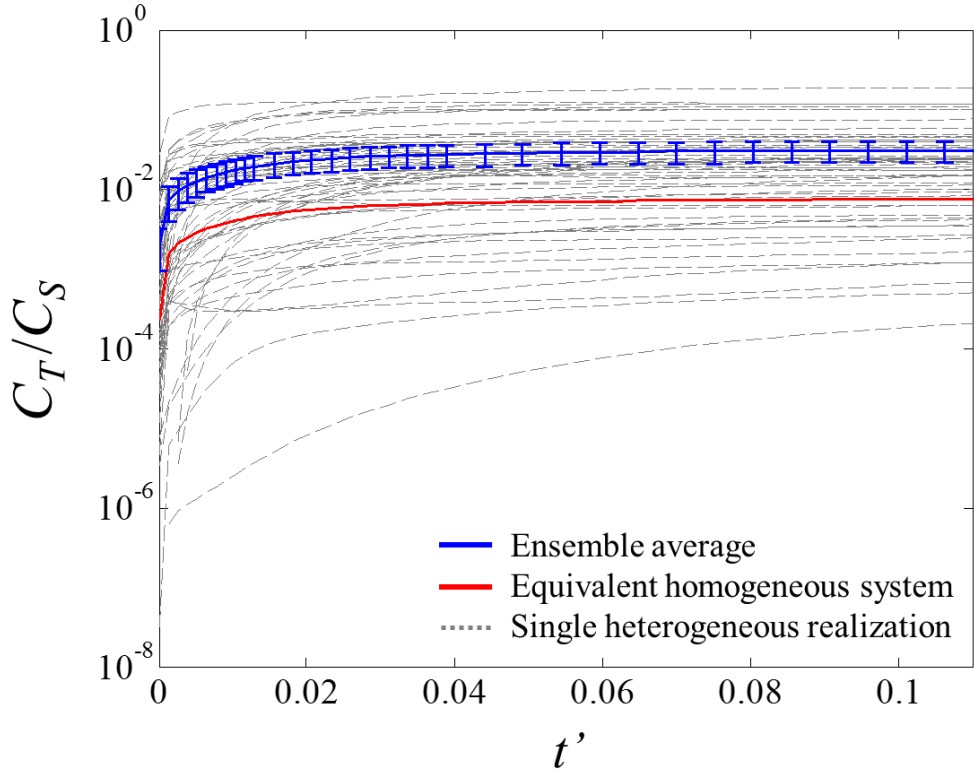

**Fig. 10.** Temporal evolution of $C_T/C_S$ for pumping scheme S3 within the equivalent homogeneous system (red curve) and for each heterogeneous realization (gray dashed curves). The ensemble average (blue curve) dimensionless concentration and its 95%-confidence intervals are also depicted.