# Peer review of "Groundwater withdrawal in randomly heterogeneous coastal aquifers"

_Hydrology and Earth System Sciences, 2017_

## Referee Comment (RC1) · Anonymous Referee #1 · 9 Aug 2017

In general, the study of heterogeneity effects on SWI is a worthwhile endeavour. Unfortunately, the investigation results are not generalizable, and where conclusions are made, they are all conclusions found in other, previous articles.

Specific comments: Page 1 Abstract: L8: "Mediterranean" isn't needed because "worldwide" includes the Mediterranean. Abstract generally: There are not new findings presented in the Abstract. The title reflects a generic investigation, whereas the Abstract describes a more site-specific investigation, but regardless, there is nothing that is new in the Abstract, because seawater pumping to reduce SWI has already been studied, as has the effect of heterogeneities on SWI. There needs to be clear guidance in the Abstract as to what is an advance on the existing body of scientific knowledge regarding this topic. Also, the Abstract reads as though a single well has been used in

studying SWI. This would be an extremely rare situation – i.e. a single well pumping. It is more likely that there are many wells being used within a coastal aquifer. The limitations of using only a single well to study SWI need to be considered. Introduction: L22: "worldwide" can be removed without losing any meaning. L22: Grammar problem – suggest "threatened by seawater intrusion (SWI), which can" L24-25: The phrase "civil purpose" is not clear. Please use a phrase that is clearer. L25: "Highly critical scenarios are associated" is awkward. Suggest something like "Critical SWI thresholds are reached when seawater reaches extraction wells..." L27: "Mas Pla" is not spelt in the same way in the references list. L29: "subordinated to" is an odd phrase to use here. "dependent on" is more accessible to the readership and clearer. Page 2 L6-7: Commas used inconsistently in the formatting of citations. Also at L16 and elsewhere in the ms. L25: There is a disjoint in the flow of this paragraph. The sentence describing Abarca et al.'s (2007) work does not follow logically from the previous sentences. L27: "rely" should be "relied" to be consistent in the use of past tense in previous sentences. Page 3 L4-13: The list of examples of field-scale SWI studies does include pivotal cases. For example, Dougeris and Zissis (2014) is a synthetic case that considers steady-state schemes, so it is hardly worth mentioning. Narayan et al. (2007) is a 2D model of a very idealised version of the field scale problem. On the other hand, Dausman and Langevin (2005; Movement of the Saltwater Interface in the Surficial Aquifer System in Response to Hydrologic Stresses and Water-Management Practices, Broward County, Florida: U.S. Geological Survey Scientific Investigations Report 2004-5256, 73 p.) and Werner and Gallagher (2006; Regional-scale, fully coupled modelling of stream-aquifer interaction in a tropical catchment, Journal of Hydrology 328: 497-510) provided early examples of comprehensive field-scale, three-dimensional SWI modelling. L18: Correct to "considering variable-density flow" L21: Correct to "spatial patterns of salt" L24: The statement about "...uncertainty in the displacement..." needs more information. What sort of uncertainty is this exactly – related to the lack of knowledge of heterogeneities or other aquifer properties? It isn't clear. L26: I don't understand what is meant by "average concentration fields", to the degree that I can't

offer possible interpretations or alternative wording. Page 4 L2: "and" needed before "(iii) reducing. . ." L12-14: Recommending deleting this last paragraph –it is not needed for journal articles. L18: "river" should be "River". Same at L22, L27 and elsewhere. L21: Correct to: "is mainly composed of a" Page 5 Section 2.1 generally: The area 2.5 km by 750 m is a small region. Why was this particular region chosen? L7: Where is states that the underlying clay acts as an impermeable barrier, is this saying that a clay sequence is presumed to represent the base of the model domain? It should be clearer. L8: "embedded" is the wrong word here. "using" or "based on" would be better. L10: "fluids" should be "fluid" L10 onwards: I won't correct any English issues from this point, but it should be pointed out that these are numerous in the remainder of the manuscript. Page 6 L1: Please provide the units for Dm L6: There is no need to redefine variables that are already defined. L8: Use a comma in "101,632" L13: The choice of longitudinal dispersivity (aL) is critical. Because the model is heterogeneous, then aL should be smaller – it otherwise seems a little on the high side. Also, the vertical aL should be smaller than the horizontal aL, otherwise, solutes move between layers too easily (i.e. given that deltaic sediments are usually layered, thereby providing more resistance to flow and transport in the vertical than in the horizontal direction). L16: The use of no-flow boundaries is concerning. Topographical divides are unlikely to be no flow boundaries at this small scale. Perhaps the no flow boundaries running perpendicular to the coast are presumed to follow flow lines, rather than topographical divides. L17: The lack of offshore extension of the coastal aquifer should be mentioned as an area of possible error. L18: I though that the inland boundary was no flow, on the basis of the previous sentences, but now it reads as though the inland boundary is a specified flux boundary. The earlier text should be clearer about which boundaries are specified as no flow boundaries. Page 7 Generally: The initial conditions are not given or explained. The time-stepping is not explained. The approach to transience is not explained. The approach to setting pumping is not explained. Page 8 Generally: The variability that has been obtained across the various realisations is entirely dependent on the assumptions about the heterogeneous K field. If different geostatistical properties were adopted, then the outcomes would be different. How can the reader connect the variability should here (i.e., in the extent of seawater) to reality? Generally – I must have missed where it states at what time the results are given – at the end of 8 years? L18-28: This is methodology and belongs in the Material and Methods section, not in the results. P8-9: I am unable to find any new outcomes, beyond those obtained from previous research, from Section 3.1. P9-10: The scenario here for pumping should have been given in the Methods section. Also, the scenario is very site specific, so it is not clear how generalizable findings can be drawn from it. P12-13, Conclusions: The conclusions don't need to restate the methodology. This is more so done in the Abstract. Regarding the conclusion points: (1) This was already known and should not be a conclusion from this research. Of course heterogeneity influences seawater extent. Also, the rotation effect was expected on the basis of previous studies. (3) I don't understand the advice given about average concentration fields. I don't know anyone who is doing this. Also, the advice given here is stated as though it can be considered generic, but it is entirely dependent on the geostatistical parameters and the field-scale case study that form the basis of the analysis. (4) All of this advice on pumping is known from previous studies, but is stated here as though it is being advised for the first time. A proper recognition of the knowledge contained in previous studies is needed to avoid giving the wrong impression that the current study was the first to make such conclusions. The references need attention – so that consistent formatting is achieved.

---

## Referee Comment (RC2) · Anonymous Referee #2 · 21 Aug 2017

GENERAL COMMENTS

Seawater intrusion is a major problem in coastal aquifers, and several studies are attempting to improve its numerical simulation. The authors want to underline how 1) the heterogeneity of the porous media impacts the numerical simulations of coastal aquifers and 2) different configurations of the pumping scheme effect the position of the saltwater wedge and the width of the mixing zone. To answer these questions, the numerical solutions of the coupled flow and transport equations are compared considering homogenous and randomly heterogeneous permeability of the porous media. I find the topic of the manuscript of interest for HESS readers. The methodology presented is clear and the manuscript is well written.

However, in my opinion further investigation is needed to better support the conclusions

[Figure]

Creative Commons CC-BY license logo

proposed. In particular I am concerned with the following points:

1) The Monte Carlo analysis is performed using only 60 random realizations. I can understand that MC simulations of this 3D, coupled system are computational intensive, however a brief analysis on the convergence of the MC scheme is required to understand the sensitivity of the first and second moments of the computed metrics to the ensemble size (e.g. in the case without pumping).

2) Most of the conclusions are not fully supported by the results, as only one aquifer and one heterogeneous configuration have been considered (e.g., the first point: 'heterogeneous aquifer systems are characterized by tow penetration and extent of the mixing zone that are respectively smaller and larger than their counterparts ...' . An analysis of the variability of the considered metrics with respect different configurations of the permeability random field (e.g. large/small variance and large/small correlation length) would better support the proposed general conclusions. Otherwise, the conclusions should be revised referring only to the case studied.

3) By considering only three pumping schemes, I find hard to conclude that the position proposed in S3 is the best. How did the authors select the position of the well in S3? Is it possible to select the position in such a way to minimise the considered metrics (e.g. for one configuration of the random permeability)?

SPECIFIC COMMENTS

Page 6, line 19: I was not able to find the reference Almagro Landò et al. (2010). Please, report in the manuscript the details about the recharge and the head in the inland. It should be stated that these boundary conditions as well as the assumption of a fully saturated domain play a fundamental role in the determination of the SWI.

Section 2.3: which are the initial conditions for the flow and concentration equations?

Section 3.1: during the 8 years of the simulation, has the recharge any impact on the SWI? Is the solution after 8 years independent from the choice of the initial conditions?

Page 8, lines 18-29: these metrics should be presented in the 'Materials and Methods' section. A table summarizing the meaning of the seven metrics could be of great help to better follow the results.

Section 3.2: the description of the four pumping schemes (S0-S3) should be presented in the 'Materials and Methods' section.

Page 13, line 11: replace 'associated a' with 'associated with a'.

Figure 1: Could you provide a small map of Spain indicating where is the Argentona aquifer? It would also help to delineate the boundary of the model grid in panel (a).

Figure 2: please indicate the depth of the left and right boundaries in panels (b) and (c).

Figure 4: the variability of the considered metrics with respect to the single random realisations is not of interest, as it is already expressed in the confidence interval associated to the ensemble mean. It would be more interesting to see their sensitivity to different parameters describing the spatial correlation of the permeability (e.g., short vs long correlation length, high vs low variance).

Figure 10: the vertical bars representing the 95 % confidence interval should be much wider. Why the authors divided the standard deviation by the square root of n (page 12, line 12)? This operation should already be done in the computation of the standard deviation. Please, check the result and correct the figure.

References: Almagro Landò et al. (2010): is this document public? This document is cited several times along the manuscript, but it seems to be not available online. Could the author upload this report?

---

## Referee Comment (RC3) · Anonymous Referee #3 · 17 Sep 2017

The authors study seawater intrusion in a three-dimensional heterogeneous aquifer using a stochastic approach. The aquifer characteristics are inspired by the Argentona aquifer in the Maresme region of Catalonia, Spain. The authors investigate the joint effect of heterogeneity and groundwater withdrawal on the width of the mixing zone and toe position (defined in terms of salt concentration isoline) of the saltwater wedge in a fully coupled variable density flow and transport scenario. First the authors consider the impact of heterogeneity on mixing zone and toe confirming the finding of previous studies. Then the effect of three different pumping scenarios is studied. Scenarios S1 and S2 pump at a location landward outside the transition zone between sea and freshwater, S3 is located within the transition zone. In S1, the well is screened in the upper part of the aquifer, for S2 and S3 and additional screen is added in the lower part

of the aquifer. It is found that S1 and S2 cause the toe to move inland and spread at the aquifer bottom while the width of the mixing zone is not affected much. For S3 in contrary, the toe location moves seaward rapidly and then stabilises. The width of the mixing zone initially increases and then decrease toward a stable value of the order of the value without pumping. It is concluded that S3 is the most efficient scenario in reducing toe penetration. Heterogeneity leads to a reduced toe penetration for S1 and S1 compared to the equivalent homogeneous scenario, while it is similar for S3. The ensemble averaged concentration field leads consistently to an overestimation of the mixing zone as observed without pumping.

The numerical Monte-Carlo analysis is sound. However, while the authors provide a thorough literature review in the Introduction, it does not become clear, which are the open questions that are addressed in the manuscript compared to the state of the art. This is of particular interest because many aspects of heterogeneity and pumping in variable density scenarios have been discussed in the literature. This is the case, for example for the effect of three-dimensional heterogeneity in Section 3.1. Thus, the authors should make an additional effort of identifying the knowledge gaps in the light of the state of the art, formulate their research objectives and clearly indicate which of their findings go beyond the state of the art.

---

## Author Comment (AC1) · 7 Oct 2017

October, 7 2017

**Re: Response to the review of Anonymous Referee #1 of the manuscript "Groundwater withdrawal in randomly heterogeneous coastal aquifers" by Martina Siena and Monica Riva.**

We are grateful to Anonymous Referee #1 for his/her careful and detailed review of our manuscript. Following, is an itemized list of his/her comments (in italic) and our responses.

1. *In general, the study of heterogeneity effects on SWI is a worthwhile endeavour. Unfortunately, the investigation results are not generalizable, and where conclusions are made, they are all conclusions found in other, previous articles. Specific comments: Page 1 Abstract: L8: "Mediterranean" isn't needed because "worldwide" includes the Mediterranean. Abstract generally: There are not new findings presented in the Abstract. The title reflects a generic investigation, whereas the Abstract describes a more site-specific investigation, but regardless, there is nothing that is new in the Abstract, because seawater pumping to reduce SWI has already been studied, as has the effect of heterogeneities on SWI. There needs to be clear guidance in the Abstract as to what is an advance on the existing body of scientific knowledge regarding this topic.*

We agree with the Reviewer on the observation that in the literature there is a quite vast (and not always original) number of contributions dealing with SWI in homogeneous and heterogeneous systems. Roughly speaking, among about 2,000 ISI papers concerning SWI have been published from the '80. Almost 100 of these considered heterogeneous systems, while only 20 included the presence of pumping wells. A quite recent review on SWI phenomena has been offered by Werner et al. (2013).

We do remark that most of the aforementioned contributions (a) deal with deterministic approaches, where the attributes of the system (e.g., permeability) are known (or determined via an inverse modeling procedure), so that (b) the impact of uncertainty of system attributes on target environmental (or engineering) performance metrics is not truly considered. This is in stark contrast with the widely documented and recognized issue that a complete knowledge of aquifer properties is not possible. This is due to a number of reasons, including data availability, i.e., available data are most often too scarce or too sparse to yield an accurate depiction of the subsurface system in all of its relevant details (e.g., Rubin, 2003). In this context, the inherent uncertainty associated with aquifer systems must be considered, this objective being achieved framing our analyses within a stochastic approach. The latter enables us not only to provide predictions of an output quantity of interest, but also to quantify the uncertainty associated with such predictions, to be used (for example) in environmental risk assessment and probabilistic management and protection protocols for water resources. Our work is precisely set in this framework.

As we mentioned above, only a few contributions studying SWI within a stochastic framework have been published to date (less than 20 ISI-ranked papers). Amongst these, the studies most relevant to our work have been listed in the Introduction of the manuscript. It has to be noted that the vast majority of these works consider idealized synthetic showcases and/or simplified systems (typically in a two-dimensional context) and/or simple flow conditions (usually steady state mean uniform flow). To the best of our knowledge, there are only two contributions where a probabilistic approach is employed to analyze the transient behavior of a real (three-dimensional) costal aquifer: (*a*) Lecca and Cau (2009), and

(*b*) Kerrou et al. (2013), respectively targeting the Oristano (Italy) and the Korba aquifer (Tunisia). In Lecca and Cau (2009), the aquifer system is modeled by considering a (homogeneous) shallow phreatic unit and a (homogeneous) deeper unit, confined by an aquitard characterized by stochastically-varying hydraulic conductivity and variable thickness. The production of freshwater is simulated at locations in the model corresponding to the position where exploitation wells operate in the field. Kerrou et al. (2013) analyze the effects of uncertainty in permeability and distribution of pumping rates on SWI. In both studies, SWI phenomena are analyzed in terms of iso-probability maps corresponding to a target concentration (equal to 0.1). Kerrou et al. (2013) evaluate the regions delimited by the 0.05 and 0.95 iso-probability lines.

In the context illustrated above, considering the very limited number of stochastic studies, we are convinced that our work is markedly relevant to show the way stochastic approaches can find their place in the assessment of real settings.

Key elements of novelty in our work include the introduction and the detailed analysis of an original set of metrics, aimed at characterizing quantitatively the effects of heterogeneity on the extent of seawater wedge penetration and of the seawater/freshwater mixing zone. These metrics yield a quantitative depiction of SWI in a global sense across a three-dimensional system (not only at the bottom of the aquifer and/or along the vertical direction, as is usually done in the literature). Additionally, the effects of pumping on SWI are investigated by comparing three diverse withdrawal scenarios. These are designed by varying the distance of the wellbore from the coastline and from the freshwater-saltwater mixing zone. While the effectiveness of simultaneous pumping of freshwater and seawater to reduce SWI has been already investigated in the literature (e.g., Aliewi et al., 2001, Pool and Carrera, 2009, and Saravanan et al., 2014; as we clearly state in Section 1 of our manuscript), it has to be noted that in this work we evaluate for the first time the effectiveness of the so-called "negative barriers" in limiting intrusion within a randomly heterogeneous aquifer.

We will further stress the aspects of novelty of our contribution in the revised manuscript.

2. *The Abstract reads as though a single well has been used in studying SWI. This would be an extremely rare situation – i.e. a single well pumping. It is more likely that there are many wells being used within a coastal aquifer. The limitations of using only a single well to study SWI need to be considered.*

The hypotheses and limitations of our work will be further clarified in the revised manuscript. Considering our replies to item 1 of the Reviewer, we do think that the joint interaction of more than one pumping well and their effect of SWI, albeit of interest, is beyond the scope of the present work and could constitute by itself the topic of a future study.

3. *Introduction: L22: "worldwide" can be removed without losing any meaning. L22: Grammar problem – suggest "threatened by seawater intrusion (SWI), which can" L24-25: The phrase "civil purpose" is not clear. Please use a phrase that is clearer. L25: "Highly critical scenarios are associated" is awkward. Suggest something like "Critical SWI thresholds are reached when seawater reaches extraction wells: :" L27: "Mas Pla" is not spelt in the same way in the references list. L29: "subordinated to" is an odd phrase to use here. "dependent on" is more accessible to the readership and clearer. Page 2 L6-7: Commas used inconsistently in the formatting of citations. Also at L16 and elsewhere in the ms. L25: There is a disjoint in the flow of this paragraph. The sentence describing Abarca et al.'s (2007)*

*work does not follow logically from the previous sentences. L27: "rely" should be "relied" to be consistent in the use of past tense in previous sentences.*

We thank the Reviewer for pointing out the aforementioned typos and misspellings. We will fix them in the revised manuscript.

4. *Page 3 L4-13: The list of examples of field-scale SWI studies does include pivotal cases. For example, Dougeris and Zissis (2014) is a synthetic case that considers steady-state schemes, so it is hardly worth mentioning. Narayan et al. (2007) is a 2D model of a very idealised version of the field scale problem. On the other hand, Dausman and Langevin (2005; Movement of the Saltwater Interface in the Surficial Aquifer System in Response to Hydrologic Stresses andWater-Management Practices, Broward County, Florida: U.S. Geological Survey Scientific Investigations Report 2004- 5256, 73 p.) and Werner and Gallagher (2006; Regional-scale, fully coupled modelling of stream-aquifer interaction in a tropical catchment, Journal of Hydrology 328: 497- 510) provided early examples of comprehensive field-scale, three-dimensional SWI modelling.*

We thank the Reviewer for the references suggested. We remark that these studies do not consider stochastic heterogeneity, which is the main driver of our work. We also note that the work of Werner et al. (2006), mentioned by the Reviewer, does not include density effects. Conversely, the work of Werner and Gallagher (2006; Characterisation of sea-water intrusion in the Pioneer Valley, Australia using hydrochemistry and three-dimensional numerical modelling, Hydrogeology Journal, 14: 1452-1469) will be referenced in our revised manuscript in the context of field-scale deterministic SWI studies.

5. *L18: Correct to "considering variable-density flow" L21: Correct to "spatial patterns of salt".*

We thank the Reviewer for pointing out these typos and misspellings. We will fix them in the revised manuscript.

6. *L24: The statement about ": : :uncertainty in the displacement: : :" needs more information. What sort of uncertainty is this exactly – related to the lack of knowledge of heterogeneities or other aquifer properties? It isn't clear. L26: I don't understand what is meant by "average concentration fields", to the degree that I can't offer possible interpretations or alternative wording.*

The ensemble average (or mean) concentration field is evaluated by averaging solute concentration across the total number of Monte Carlo realizations. It is a function of space and time. The uncertainty associated with the system behaviour stems from the random nature of the permeability field. We will make this point unambiguously clear in the revised manuscript.

7. *Page 4 L2: "and" needed before"(iii) reducing: : :" L12-14: Recommending deleting this last paragraph –it is not needed for journal articles. L18: "river" should be "River". Same at L22, L27 and elsewhere. L21: Correct to: "is mainly composed of a"*

We apologize and we will correct those typos. Lines 12-14 describe the organization of the manuscript. Such a description is included in many papers, across a variety of Journals (including HESS). We will abide by the Editor's decision on this issue.

8. *Page 5 Section 2.1 generally: The area 2.5 km by 750 m is a small region. Why was this particular region chosen?*

As we mention in the manuscript, we started by considering the two-dimensional constant-density model developed by Rodriguez Fernandez (2015) over the whole river basin (see Fig. 1a of the manuscript). First, we developed a three-dimensional (variable-density) model on an area of 2.5 km (along the coast) by 2 km (inland). A series of preliminary numerical simulations were performed considering a homogenous domain as well as analyzing a limited number of random realizations. Results indicated that values of salt concentration at the end of the 8-year period were appreciable only in a narrow (less than 400m-wide) region close to the coast. On the basis of such preliminary runs, we designed the size of the study area analyzed in the manuscript. This choice has the additional advantage of enabling us to set up a refined computational grid towards the sea boundary (where density-driven effects are relevant) while keeping an affordable computational cost. Note that, the selected model requires about 3 hours of CPU time on a single i7-3930K Intel core provided with 32GB memory for each MC simulation. We will add some details about this issue in the revised manuscript.

9. *L7: Where is states that the underlying clay acts as an impermeable barrier, is this saying that a clay sequence is presumed to represent the base of the model domain? It should be clearer.*

Yes, it is correct. We will make this point unambiguous in the revised manuscript.

10. *L8: "embedded" is the wrong word here. "using" or "based on" would be better. L10: "fluids" should be "fluid".*

We thank the Reviewer for pointing out the aforementioned typos and misspellings.

11. *Page 6 L1: Please provide the units for Dm L6: There is no need to redefine variables that are already defined. L8: Use a comma in "101,632" L13: The choice of longitudinal dispersivity (aL) is critical. Because the model is heterogeneous, then aL should be smaller – it otherwise seems a little on the high side. Also, the vertical aL should be smaller than the horizontal aL, otherwise, solutes move between layers too easily (i.e. given that deltaic sediments are usually layered, thereby providing more resistance to flow and transport in the vertical than in the horizontal direction).*

According to Eq. (4), the units of $D_m$ are the same as those of $\boldsymbol{D}$, introduced at line 23, page 5. As discussed in the manuscript (page 6 lines 12-13), the value of longitudinal dispersivity, $\alpha_L$, has been chosen such as $\Lambda \leq 4\alpha_L$, $\Lambda$ being the element length measured along the direction of flow (Voss, 1984), to ensure stability. The adoption of smaller $\alpha_L$ would led to unfeasible computational times. For the purpose of our simulations, transverse horizontal and vertical dispersivity values have been set one order of magnitude smaller than the longitudinal dispersivity, i.e., $\alpha_T = 0.5$ m in our work, as it is commonly assumed in the literature (e.g., Cobaner et al., 2012; Koussis et al., 2002; Narayan et al., 2007).

*12. L16: The use of no-flow boundaries is concerning. Topographical divides are unlikely to be no flow boundaries at this small scale. Perhaps the no flow boundaries running perpendicular to the coast are presumed to follow flow lines, rather than topographical divides. L17: The lack of offshore extension of the coastal aquifer should be mentioned as an area of possible error.*

We agree with the Reviewer in that topographic divides and groundwater divides may not coincide. We started by identifying lateral boundaries as groundwater divides consistent with the previous two-dimensional model that was set up for the area (Rodriguez Fernandez, 2015). A close inspection of Figs. 1b and 3 reveals that these no-flow boundaries are indeed (approximately) perpendicular to the coastline in the portion of the domain here considered. We will clarify this point in the revised manuscript. We will also mention that the offshore extension is neglected in the current investigation.

*13. L18: I though that the inland boundary was no flow, on the basis of the previous sentences, but now it reads as though the inland boundary is a specified flux boundary. The earlier text should be clearer about which boundaries are specified as no flow boundaries.*

The inland boundary condition set in our model is defined for the first time at the point noted by the Reviewer. We do think this to be the appropriate place to specify all inputs (including boundary conditions) of our model.

*14. Page 7. The initial conditions are not given or explained. The time-stepping is not explained. The approach to transience is not explained. The approach to setting pumping is not explained.*

We set $h = 0$ as initial condition. Adopting $h = 2.4$ m (equal to the mean value of $h$ set along the inland boundary) did not lead to significantly diverse results at the end of the 8-year time period in the homogeneous system. As it is commonly done in the literature, (e.g., Bear et al., 2001; Koussis et al., 2002; Jakovovic et al., 2016) we set initially $C = C_F = 0$. A uniform time step $\Delta t = 1$ day has been set during the 8-year run. A higher time resolution was required for the subsequent 8-month period, due to the activation of pumping. We then set $\Delta t = 30$ min for the first month and progressively increased the time step, up to $\Delta t = 120$ min, as the system showed smoother variations while approaching steady-state. Pumping is implemented by setting a flux-type condition in all cells included in the well-screens. The total rate $Q$ extracted is uniformly distributed across the numerical blocks associated with the screen.

*15. Page 8 The variability that has been obtained across the various realisations is entirely dependent on the assumptions about the heterogeneous K field. If different geostatistical properties were adopted, then the outcomes would be different. How can the reader connect the variability should here (i.e., in the extent of seawater) to reality?*

The adoption of diverse geostatistical models (e.g., non Gaussian distribution of $k$, conceptualization of the system as a composite medium, or others) would probably lead to different results. Albeit of interests, the analysis of the effects of diverse geostatistical models on SWI metrics is outside the scope of the current contribution. With reference to the type of random heterogeneity analyzed, we note that spherical variograms have been employed to describe a variety of field settings.

The variogram sill we consider represents a domain with moderate variability. As we state in the manuscript, the value of the correlation scale has been selected consistent with documented analyses according to which the integral scale of log conductivity and transmissivity values inferred worldwide using traditional (such as exponential and spherical) variograms tends to increase with the length scale of the sampling window at a rate of about 1/10 (Gelhar, 1993; Neuman et al., 2008). We remark that ours is one of the first attempts at including the effect of random heterogeneity within a three-dimensional, transient density-variable system (see also our answer to item 1). We concur that a systematic analysis of the influence of variogram shape and variogram parameter values would be of interest and will be the subject of a future study.

16. *Page 8 L18-28: This is methodology and belongs in the Material and Methods section, not in the results.*

We will move this part to the Material and Methods section in the revised manuscript.

17. *P8-9: I am unable to find any new outcomes, beyond those obtained from previous research, from Section 3.1.*

Please see our answer to item 1.

18. *P9-10: The scenario here for pumping should have been given in the Methods section. Also, the scenario is very site specific, so it is not clear how generalizable findings can be drawn from it.*

We will move this part to the Material and Methods section in the revised manuscript. We investigate the effects of pumping on SWI by comparing three diverse withdrawal scenarios, designed by varying the screen location along the vertical direction and the distance of the wellbore from the coastline and from the freshwater-saltwater mixing zone. In this work we evaluate the effectiveness of each pumping scenario within a randomly heterogeneous system in terms of local and global metrics describing the extent of seawater intrusion and salt mass fraction at the freshwater well (see also our answer to item 1). We are aware that our study does not cover the totality of feasible combinations of pumping scenarios and heterogeneity. As already remarked, this is the first attempt to include the effect of (*i*) random heterogeneity and (*ii*) simultaneous extraction of freshwater and seawater within a three-dimensional (variable-density) realistic system.

19. *P12-13, Conclusions: (1) This was already known and should not be a conclusion from this research. Of course heterogeneity influences seawater extent. Also, the rotation effect was expected on the basis of previous studies.*

We partially agree with the Reviewer's comment. Heterogeneity effects have been already observed in deterministic models, or in stochastic analyses invoking ergodicity assumption. We will revise the conclusions highlighting the novel elements of our study, as we detail in our reply to item 1.

20. *(3) I don't understand the advice given about average concentration fields. I don't know anyone who is doing this. Also, the advice given here is stated as though it can be*

*considered generic, but it is entirely dependent on the geostatistical parameters and the field-scale case study that form the basis of the analysis.*

As we mentioned in Section 1 of the manuscript (page 3, line 23), in several works (e.g., Rivest et al., 2012; Lu et al., 2016) concentration values at unsampled locations are obtained by applying a kriging procedure on the basis of available concentration data. Since Kriging is an estimation method, it provides an estimate of the concentration. Therefore, the *kriged* value of concentration should be regarded as the mean (or average) value of an otherwise random concentration. For this reason, we state that the authors mentioned above used "average concentration fields" and such fields should be compared against our "Ensemble-averaged concentration" field (see also our answer to item 6 for the definition of ensemble-averaged concentration). Possible limitations of our study, in terms of the geostatistical model and associated parameter values adopted, will be further stressed in the revised manuscript.

21. *(4) All of this advice on pumping is known from previous studies, but is stated here as though it is being advised for the first time. A proper recognition of the knowledge contained in previous studies is needed to avoid giving the wrong impression that the current study was the first to make such conclusions.*

Some of these conclusions were already associated with previous deterministic (homogeneous and heterogeneous) models. Here, we are pointing out that these are indeed valid within a stochastic framework. We will revise this conclusion to stress this important issue.

22. *The references need attention so that consistent formatting is achieved.*

Many thanks. We will update the format of the references in the revised manuscript.

**References (note that only the references not already included in the manuscript are listed)**

Bear, J., Zhou, Q., Bensabat, J.: Three dimensional simulation of seawater intrusion in heterogeneous aquifers, with application to the coastal aquifer of Israel. First International Conference on Saltwater Intrusion and Coastal Aquifers – Monitoring, Modelling and Management, April 23–25, Essaouira, Morocco, 2001.

Dausman, A. and Langevin, C.D.: Movement of the Saltwater Interface in the Surficial Aquifer System in Response to Hydrologic Stresses and Water-Management Practices, Broward County, Florida, U.S. Geological Survey Scientific Investigations Report 2004- 5256, 73 p., 2005.

Kerrou, J., Renard, P., Cornaton, F. and Perrochet, P., Stochastic forecasts of seawater intrusion towards sustainable groundwater management: application to the Korba aquifer (Tunisia), Hydrogeology J., 21, 425-440, doi: 10.1007/s10040-012-0911-x, 2013.

Jakovovic, D., Werner, A. D., de Louw, P. G. B., Post, V. E. A., Morgan, L. K., Saltwater upconing zone of influence, Adv. Water Resour., 94, 75-86, doi: 10.1016/j.advwatres.2016.05.003, 2016.

Lecca, G., and Cau, P., Using a Monte Carlo approach to evaluate seawater intrusion in the Oristano coastal aquifer: A case study from the AQUAGRID collaborative computing platform, Phys. Chem. Earth, 34, 654-661, 2009.

Rubin, Y., Applied stochastic hydrogeology. New York, NY: Oxford University Press, 2003.

Voss, C., A Finite-Element Simulation Model for Saturated-Unsaturated, Fluid-Density-Dependent Groundwater Flow with Energy Transport or Chemically-Reactive Single-Species Solute Transport, Ann. Phys. (N.Y.), 54, 258, 1984.

Werner, A. D., Gallagher, M. R., and Weeks, S. W., Regional-scale, fully coupled modelling of stream-aquifer interaction in a tropical catchment, J. Hydrol. 328, 497-510, doi:10.1016/j.jhydrol.2005.12.034, 2006.

Werner, A. D. and Gallagher M. R., Characterisation of sea-water intrusion in the Pioneer Valley, Australia using hydrochemistry and three-dimensional numerical modelling, Hydrogeology J., 14, 1452-1469, doi: 10.1007/s10040-006-0059-7, 2006.

Werner, A. D., Bakker, M., Post, V. E. A., Vandenbohede, A., Lu, C., Ataie-Ashtiani, B., Simmons, C. T. and Barry, D. A, Seawater intrusion processes, investigation and management: recent advances and future challenges, Adv. Water Resour. 51 (1), 3-26, doi:10.1016/j.advwatres.2012.03.004, 2013.

---

## Author Comment (AC2) · 7 Oct 2017

October, 7 2017

**Re: Response to the review of Anonymous Referee #2 of the manuscript "Groundwater withdrawal in randomly heterogeneous coastal aquifers" by Martina Siena and Monica Riva.**

We appreciate the efforts Anonymous Referee #2 has invested in our manuscript and we are grateful for his/her insightful comments. Following is an itemized list of his/her comments (in italic) and our responses.

> *GENERAL COMMENTS*
> *Seawater intrusion is a major problem in coastal aquifers, and several studies are attempting to improve its numerical simulation. The authors want to underline how 1) the heterogeneity of the porous media impacts the numerical simulations of coastal aquifers and 2) different configurations of the pumping scheme effect the position of the saltwater wedge and the width of the mixing zone. To answer these questions, the numerical solutions of the coupled flow and transport equations are compared considering homogenous and randomly heterogeneous permeability of the porous media. I find the topic of the manuscript of interest for HESS readers. The methodology presented is clear and the manuscript is well written.*

We thank the Reviewer for his/her appreciation of our work.

> *1. However, in my opinion further investigation is needed to better support the conclusions proposed. In particular I am concerned with the following points: 1) The Monte Carlo analysis is performed using only 60 random realizations. I can understand that MC simulations of this 3D, coupled system are computational intensive, however a brief analysis on the convergence of the MC scheme is required to understand the sensitivity of the first and second moments of the computed metrics to the ensemble size (e.g. in the case without pumping).*

We agree with the Reviewer in that the number of Monte Carlo (MC) realizations, $n$, is critical. The choice of $n$ typically results from a trade-off between CPU time and accuracy in reproducing the statistical moments of the quantities of interest. For example, Pool et al. (2015) rely on $n = 50$ to analyze the effect of tidal fluctuations on SWI in a three-dimensional heterogeneous system. Additionally, the choice of the number of Monte Carlo realizations depends on the type of quantity that one is interested in (for example, either point-/local- or integral- quantities, as also detailed below) and on the type of behavior which is intended to be highlighted. Note that, for the test cases analyzed in the manuscript, a single MC simulation on a i7-3930K Intel core with 32GB memory requires about 3 hours.

In the following we assess the stability of the MC-based results on the basis of the methodology proposed by Ballio and Guadagnini (2004). Figures R1 and R2 respectively depict the sample mean and standard deviation of all metrics, $\xi''$, considered in the manuscript versus $n$. Results are evaluated (for the test case without pumping) at the end of the 8-year period considered. The estimated 95%-confidence intervals, computed according to eqs (3) and (8) of Ballio and Guadagnini (2004), are also shown (see also our reply to item 8). Figures R1 and R2 show that the oscillations displayed by the quantities of interest are in general limited and do not hamper the strength of the main massage of our work. As expected, integral quantities (i.e., $A_T''$, $A_S''$, $V_T''$, $V_S''$, $W_{MZ}''$) tend to stabilize faster than local ones ($L_T''$ and $L_S''$). We plan to (*i*) increase $n$ up to 100 (for each scenario investigated) and (*ii*) include this convergence analysis as supplementary material in the revised manuscript.

[Figure]

Figure R1. Sample mean of metrics $\xi''$ versus the ensemble size. The associated 95% confidence intervals are also shown.

[Figure]

Figure R2. Sample standard deviation of metrics $\xi''$ versus the ensemble size. The associated 95% confidence intervals are also shown.

> *2. Most of the conclusions are not fully supported by the results, as only one aquifer and one heterogeneous configuration have been considered (e.g., the first point: 'heterogeneous aquifer systems are characterized by toe penetration and extent of the mixing zone that are respectively smaller and larger than their counterparts...'. An analysis of the variability of the considered metrics with respect different configurations of the permeability random field (e.g. large/small variance and large/small correlation length) would better support the proposed general conclusions. Otherwise, the conclusions should be revised referring only to the case studied.*

The present work aims at investigating the effect of random heterogeneity on a three-dimensional domain patterned after a real aquifer. As discussed in the manuscript, only a few contributions studying SWI within a stochastic framework have been published to date. Amongst these, the studies most relevant to our work have been listed in the Introduction of the manuscript. It has to be noted that the vast majority of these works consider idealized synthetic showcases and/or simplified systems (typically in a two-dimensional context) and/or simple flow conditions (usually steady state mean uniform flow). To the best of our knowledge, there are only two contributions where a probabilistic approach is employed to analyze the transient behavior of a real (three-dimensional) costal aquifer: (*a*) Lecca and Cau (2009), and (*b*) Kerrou et al. (2013), respectively targeting the Oristano (Italy) and the Korba aquifer (Tunisia). Key elements of novelty of our manuscript with respect to these works include the introduction and the detailed analysis of an original set of metrics, aimed at characterizing quantitatively the effects of heterogeneity on the extent of seawater wedge penetration and of the seawater/freshwater mixing zone.

These metrics yield a quantitative depiction of SWI in a global sense across a three-dimensional system (not only at the bottom of the aquifer and/or along the vertical direction, as is usually done in the literature).

With reference to the type of random heterogeneity analyzed, we note that the variogram sill we consider represents a domain with moderate variability. As we state in the manuscript, the value of the correlation scale has been selected consistent with documented analyses according to which the integral scale of log conductivity and transmissivity values inferred worldwide using traditional (such as exponential and spherical) variograms tends to increase with the length scale of the sampling window at a rate of about 1/10 (Gelhar, 1993; Neuman et al., 2008). We concur that a systematic analysis of the influence of variogram shape and variogram parameter values would be of interest and will be the subject of a future study.

We stress that ours is one of the first attempts at including the effect of random heterogeneity within a three-dimensional, transient density-variable system. In this context, our results can be considered as exemplary for the type of representative field conditions we analyze. We will revise the conclusions highlighting the novel elements of our study.

> *3. By considering only three pumping schemes, I find hard to conclude that the position proposed in S3 is the best. How did the authors select the position of the well in S3? Is it possible to select the position in such a way to minimise the considered metrics (e.g. for one configuration of the random permeability)?*

The three pumping schemes have been selected to investigate the effects of the distance of the wellbore from the coastline and from the freshwater-saltwater mixing zone on SWI. In this context, in S2 and S3 we also analyzed the impact of an additional pumping rate of seawater (at the bottom of the aquifer). It has to be noted that in this work we evaluate for the first time the effectiveness of the simultaneous extraction of fresh- and seawater in limiting SWI intrusion within a three-dimensional random heterogeneous aquifer. We are aware that our study does not cover the totality of feasible combinations of pumping scenarios. The analysis proposed by Reviewer to identify the optimal well location, albeit of interest, is beyond the scope of the present work and could constitute by itself the topic of a future study. At the same time, we are convinced that such a study should be performed within a stochastic framework (not in a single realization context), thus requiring a remarkable (and possibly prohibitive, in case one would also consider multiple variogram parameters and functional shapes) CPU time.

> *4. Page 6, line 19: I was not able to find the reference Almagro Landò et al. (2010). Please, report in the manuscript the details about the recharge and the head in the inland. It should be stated that these boundary conditions as well as the assumption of a fully saturated domain play a fundamental role in the determination of the SWI.*

The area of interest is characterized by 5 recharge zones (see Fig. 1 of the manuscript) specified on the basis of the land use (inferred from the SIGPAC2005 dataset). Rodriguez Fernandez (2015) provides calibrated values of the recharge associated with each zone. The total recharge slightly varies in time (on a monthly basis), with mean value equal to about 7.6 l/s. Head values at the inland boundary are inferred by interpolating the time-dependent hydraulic-head distribution taken from the two-dimensional model of Rodriguez Fernandez (2015). It has to be noted that iso-potential curves are approximately parallel to the coast in the region of interest. Therefore, the head values set along the inland boundary are approximately constant in time. The average value of hydraulic head at the inland boundary over the 8-year period is $h = 2.4$ m.

*5. Section 2.3: which are the initial conditions for the flow and concentration equations? Section 3.1: during the 8 years of the simulation, has the recharge any impact on the SWI? Is the solution after 8 years independent from the choice of the initial conditions?*

We set $h = 0$ as initial condition. Adopting $h = 2.4$ m (equal to the mean value of $h$ set along the inland boundary) did not lead to significantly diverse results at the end of the 8-year time period in the homogeneous system. As it is commonly done in the literature, (e.g., Bear et al., 2001; Koussis et al., 2002; Jakovovic et al., 2016) we set initially $C = C_F = 0$. The impact of recharge on SWI has not been investigated. However, due to the limited recharge in the investigated area, its effect appears to be negligible. This can be inferred, for example, from Fig. 3 of the manuscript. The isolines $C/C_s = 0.5$ for the homogeneous system (red curves) do not change appreciably amongst different cross sections along the coast (characterized by diverse recharges).

*6. Page 8, lines 18-29: these metrics should be presented in the 'Materials and Methods' section. A table summarizing the meaning of the seven metrics could be of great help to better follow the results. Section 3.2: the description of the four pumping schemes (S0-S3) should be presented in the 'Materials and Methods' section. Page 13, line 11: replace 'associated a' with 'associated with a'. Figure 2: please indicate the depth of the left and right boundaries in panels (b) and (c). Figure 1: Could you provide a small map of Spain indicating where is the Argentona aquifer? It would also help to delineate the boundary of the model grid in panel (a).*

We agree with the Reviewer's suggestions and we will implement them in the revised manuscript.

*7. Figure 4: the variability of the considered metrics with respect to the single random realisations is not of interest, as it is already expressed in the confidence interval associated to the ensemble mean. It would be more interesting to see their sensitivity to different parameters describing the spatial correlation of the permeability (e.g., short vs long correlation length, high vs low variance).*

We prefer plotting not only the ensemble results but also the single realization outcomes. Please see also our answer to item 2.

*8. Figure 10: the vertical bars representing the 95 % confidence interval should be much wider. Why the authors divided the standard deviation by the square root of n (page 12, line 12)? This operation should already be done in the computation of the standard deviation. Please, check the result and correct the figure.*

Vertical bars in Figure 10 represent an estimate of the error in the evaluation of the sample mean $\langle C_T \rangle / C_S$, due to the finite value $n$ of Monte Carlo (MC) simulations. The error in the evaluation of the sample mean scales with $1/\sqrt{n}$. Assuming that $C_T$ has a Normal distribution, we can write (see e.g., Ballio and Guadagnini 2004)

$$Pr\left[\overline{\mathfrak{R}_n} - t_{n-1}\left(1 - \frac{\alpha}{2}\right)\frac{S_n}{\sqrt{n}} \leq \mu \leq \overline{\mathfrak{R}_n} + t_{n-1}\left(1 - \frac{\alpha}{2}\right)\frac{S_n}{\sqrt{n}}\right] = 1 - \alpha \tag{R1}$$

where $\mu$ is the ensemble mean of $C_T/C_S$, $\overline{\mathfrak{R}_n}$ is the sample mean (computed on the basis of $n$ realizations, denoted $\langle C_T \rangle / C_S$ in the manuscript), $S_n$ is the sample standard deviation, $t_{n-1}$ is the Student distribution with ($n$-1) degree of freedom and $1 - \alpha$ is the probability that $\mu$ lies within the confidence intervals around the sample mean $\overline{\mathfrak{R}_n}$. When $n$ is large (about $n > 30$), $t_{n-1}$ can be approximated by a

standard normal distribution. Therefore, setting $\alpha = 0.05$, the 95% confidence intervals are given by

$$\overline{\Re}_n - 1.96 \frac{S_n}{\sqrt{n}} \leq \mu \leq \overline{\Re}_n + 1.96 \frac{S_n}{\sqrt{n}}.$$

*9. References: Almagro Landò et al. (2010): is this document public? This document is cited several times along the manuscript, but it seems to be not available online. Could the author upload this report?*

The references concerning the preliminary model of the Argentona basin will be made available if needed.

**References: (note that only the references not already included in the manuscript are listed)**

Ballio, F., and Guadagnini, A., Convergence assessment of numerical Monte Carlo simulations in groundwater hydrology, Water Resour. Res., 40, 4, W04603, doi: 10.1029/2003WR002876, 2004.

Bear, J., Zhou, Q., Bensabat, J.: Three dimensional simulation of seawater intrusion in heterogeneous aquifers, with application to the coastal aquifer of Israel. First International Conference on Saltwater Intrusion and Coastal Aquifers – Monitoring, Modelling and Management, April 23–25, Essaouira, Morocco, 2001.

Kerrou, J., Renard, P., Cornaton, F. and Perrochet, P., Stochastic forecasts of seawater intrusion towards sustainable groundwater management: application to the Korba aquifer (Tunisia), Hydrogeology J., 21, 425-440, doi: 10.1007/s10040-012-0911-x, 2013.

Jakovovic, D., Werner, A. D., de Louw, P. G. B., Post, V. E. A., Morgan, L. K., Saltwater upconing zone of influence, Adv. Water Resour., 94, 75-86, doi: 10.1016/j.advwatres.2016.05.003, 2016.

Lecca, G., and Cau, P., Using a Monte Carlo approach to evaluate seawater intrusion in the Oristano coastal aquifer: A case study from the AQUAGRID collaborative computing platform, Phys. Chem. Earth, 34, 654-661, 2009.

---

## Author Comment (AC3) · 7 Oct 2017

October, 7 2017

**Re: Response to the review of Anonymous Referee #3 of the manuscript "Groundwater withdrawal in randomly heterogeneous coastal aquifers" by Martina Siena and Monica Riva.**

We appreciate the efforts Anonymous Referee #3 has invested in our manuscript and we are grateful for his/her insightful comments.

> *The authors study seawater intrusion in a three-dimensional heterogeneous aquifer using a stochastic approach. The aquifer characteristics are inspired by the Argentona aquifer in the Maresme region of Catalonia, Spain. The authors investigate the joint effect of heterogeneity and groundwater withdrawal on the width of the mixing zone and toe position (defined in terms of salt concentration isoline) of the saltwater wedge in a fully coupled variable density flow and transport scenario. First the authors consider the impact of heterogeneity on mixing zone and toe confirming the finding of previous studies. Then the effect of three different pumping scenarios is studied. Scenarios S1 and S2 pump at a location landward outside the transition zone between sea and freshwater, S3 is located within the transition zone. In S1, the well is screened in the upper part of the aquifer, for S2 and S3 and additional screen is added in the lower part of the aquifer. It is found that S1 and S2 cause the toe to move inland and spread at the aquifer bottom while the width of the mixing zone is not affected much. For S3 in contrary, the toe location moves seaward rapidly and then stabilises. The width of the mixing zone initially increases and then decrease toward a stable value of the order of the value without pumping. It is concluded that S3 is the most efficient scenario in reducing toe penetration. Heterogeneity leads to a reduced toe penetration for S1 and S1 compared to the equivalent homogeneous scenario, while it is similar for S3. The ensemble averaged concentration field leads consistently to an overestimation of the mixing zone as observed without pumping. The numerical Monte-Carlo analysis is sound.*

We thank the Reviewer for his/her appreciation of our work.

> *However, while the authors provide a thorough literature review in the Introduction, it does not become clear, which are the open questions that are addressed in the manuscript compared to the state of the art. This is of particular interest because many aspects of heterogeneity and pumping in variable density scenarios have been discussed in the literature. This is the case, for example for the effect of three-dimensional heterogeneity in Section 3.1. Thus, the authors should make an additional effort of identifying the knowledge gaps in the light of the state of the art, formulate their research objectives and clearly indicate which of their findings go beyond the state of the art.*

We thank the Reviewer for his/her very interesting and challenging comments. In the literature there is a quite vast number of contributions dealing with SWI in homogeneous and heterogeneous systems. Most

of these contributions (a) deal with deterministic approaches, where the attributes of the system (e.g., permeability) are known (or determined via an inverse modeling procedure), so that (b) the impact of uncertainty of system attributes on target environmental (or engineering) performance metrics is not truly considered. This is in stark contrast with the widely documented and recognized issue that a complete knowledge of aquifer properties is not possible. In this context, the inherent uncertainty associated with aquifer systems must be considered, this objective being achieved framing our analyses within a stochastic approach. The latter enables us not only to provide predictions of an output quantity of interest, but also to quantify the uncertainty associated with such predictions, to be used (for example) in environmental risk assessment and probabilistic management and protection protocols for water resources. Our work is precisely set in this framework. Only a few contributions studying SWI within a stochastic framework have been published to date (less than 20 ISI-ranked papers). Amongst these, the studies most relevant to our work have been listed in the Introduction of the manuscript. It has to be noted that the vast majority of these works consider idealized synthetic showcases and/or simplified systems (typically in a two-dimensional context) and/or simple flow conditions (usually steady state mean uniform flow). In this context, considering the very limited number of stochastic studies, we are convinced that our work is markedly relevant to show the way stochastic approaches can find their place in the assessment of real settings. Key elements of novelty in our work include the introduction and the detailed analysis of an original set of metrics, aimed at characterizing quantitatively the effects of heterogeneity on the extent of seawater wedge penetration and of the seawater/freshwater mixing zone. These metrics yield a quantitative depiction of SWI in a global sense across a three-dimensional system (not only at the bottom of the aquifer and/or along the vertical direction, as is usually done in the literature). Additionally, the effects of pumping on SWI are investigated by comparing three diverse withdrawal scenarios. These are designed by varying the distance of the wellbore from the coastline and from the freshwater-saltwater mixing zone. While the effectiveness of simultaneous pumping of freshwater and seawater to reduce SWI has been already investigated in the literature (e.g., Aliewi et al., 2001, Pool and Carrera, 2009, and Saravanan et al., 2014; as we clearly state in Section 1 of our manuscript), it has to be noted that in this work we evaluate for the first time the effectiveness of the so-called "negative barriers" in limiting intrusion within a randomly heterogeneous aquifer. We will further stress the aspects of novelty of our contribution in the revised manuscript.

---

## Author Response (AR1)

POLITECNICO DI MILANO
**Dipartimento di Ingegneria Civile e Ambientale**

Milano, January 19th 2018

To:
Professor Bill Hu
Editor-in-Chief
Hydrology and Earth System Sciences

Re: Submission of the revised manuscript "Groundwater withdrawal in randomly heterogeneous coastal aquifers"
by M. Siena[1] and M. Riva[1,2]

[1]Dipartimento di Ingegneria Civile e Ambientale, Politecnico di Milano, Milano, 20133, Italy.
[3]Department of Hydrology and Atmospheric Sciences, University of Arizona, Tucson, AZ 85721, USA

Dear Editor:
We appreciate the efforts you and the Referees have invested in our manuscript, and the Referees' suggestions for improvement. We are pleased to submit a revised version in response to the Reviewers' insightful comments. Modifications to the original manuscript are reported in red font.

Sincerely,
Martina Siena, Monica Riva